# Learning Search Space Boundaries Improves Supernet Training

## Abstract

Neural architecture search (NAS) seeks to automate neural network design to optimize performance criteria, but designing a search space for NAS largely remains a manual effort. When available, strong prior knowledge can be used to construct small search spaces, but using such spaces inevitably limits the flexibility of NAS, and prior information is not always available on novel tasks and/or architectures. On the other hand, many NAS methods have been shown to be sensitive to the choice of search space and struggle when the search space is not sufficiently refined. To address this problem, we propose a differentiable technique that learns a policy to refine a broad initial search space during supernet training. Our proposed solution is orthogonal to almost all existing improvements to NAS pipelines, is largely search space-agnostic, and incurs little additional overhead beyond standard supernet training. Despite its simplicity, we show that on tasks without strong priors, our solution consistently discovers performant subspaces within an initially large, complex search space (where even the state-of-the-art methods underperform), significantly robustifies the resultant supernet and improves the performance across a wide range model sizes. We argue that our work takes a step toward full automation of the network design pipeline.

## 1 Introduction

Over the last half-decade, neural architecture search (NAS), which aims to automate the design of neural network architectures for various tasks, has seen great successes: for example, in a wide range of tasks (Zoph et al., 2018; Chen et al., 2019; Liu et al., 2019a; Zhang et al., 2019), architectures designed by NAS often outperform handcrafted networks designed by human experts. Many early NAS methods adopt a *query-based* paradigm by repeatedly training and refining models via, for example, reinforcement learning (RL) (Zoph et al., 2018; Tan et al., 2019; Baker et al., 2017)), evolutionary algorithms (Real et al., 2019; Liu et al., 2021; Real et al., 2017) and/or Bayesian optimization/quadrature (White et al., 2021; Ru et al., 2021; Wan et al., 2022a; Kandasamy et al., 2018; Hamid et al., 2023); these methods typically require prohibitive amounts of computing resources even on simple vision tasks (e.g., early RL-based methods require thousands of GPU-hours even on simple CIFAR tasks). More recent methods typically leverage *weight-sharing supernets* to conduct architecture search in a one-shot manner without training candidate architectures individually (Brock et al., 2018; Pham et al., 2018; Liu et al., 2019b; Guo et al., 2020; Li & Talwalkar, 2020; Cai et al., 2019): typically, modern NAS methods first train *all* candidate networks in the search space $\mathcal{A}$ via parameter-sharing supernets (*supernet training*), which is followed by *architecture selection* to identify the most promising candidate sub-networks that lead to best trade-offs between performance and costs (e.g., in terms of model size, latency, etc.). While earlier supernet-based methods often require re-training the resulting sub-networks (Pham et al., 2018; Liu et al., 2019b) or at least fine-tuning (Cai et al., 2020), more recent advancements, such as the use of *in-place knowledge distillation* and *sandwich sampling* proposed by Yu & Huang (2019) and subsequently widely used in related literature (Yu et al., 2020; Wang et al., 2021a; Gong et al., 2022; Wang et al., 2021b), have enabled practitioners to obtain performant sub-networks by simply slicing the supernet appropriately without further training (Yu et al., 2019).

Although these advances in the search methodology have democratized the usage of NAS by reducing its computational costs, choosing a good *search space* remains a practical challenge that has received less attention.

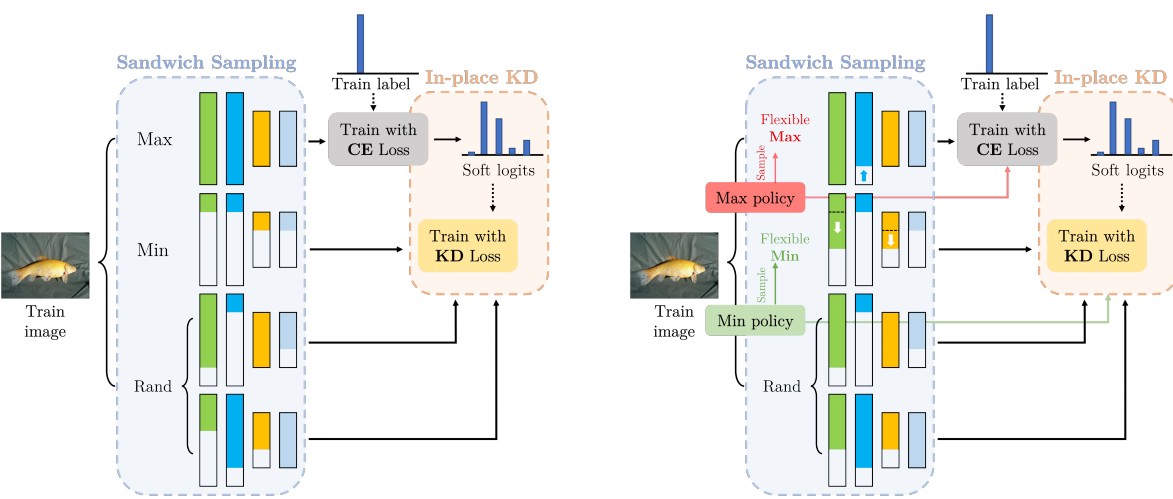

Figure 1: Illustration of an iteration in baseline supernet training (§2, **left**) and training with boundary learning (§3, **right**): at each mini-batch, the max and min networks are always sampled alongside random networks from $\mathcal{A}$ (sandwich sampling), and only the max network is trained with cross-entropy loss with the true label; the other networks are trained with knowledge distillation using the soft logits of the max network as the teacher (in-place KD). In baseline supernets (**left**), the max and min networks are fixed to be the largest and smallest sub-networks in the search space. In contrast, we propose to learn max and min *policies* and *sample* max and min networks from those policies at each iteration to use as the endpoints in sandwich sampling. Importantly, the sampled max and min networks may differ from the ground-truth max and min networks, respectively, and the policies are optimized in an end-to-end manner jointly with the supernet training losses (CE and KD losses) (**right**).

While large and expressive search spaces are, in principle, desirable or even crucial for discovering high-performing and novel architectures, it has been shown that NAS methods, particularly the supernet-based methods, require meticulously designed search spaces. Previous works have shown that NAS performance is sensitive to search space design—arguably more so than the choice of search algorithm (Wan et al., 2022b; Yang et al., 2020). Moreover, directly applying many existing popular NAS methods in large, unrefined search spaces has been observed to lead to sub-optimal performance (Ci et al., 2021; Zela et al., 2020)—and as we will show, such issues persist even with modern, state-of-the-art search methods. As a result, many works benchmark methods on carefully engineered search spaces often constructed by modifying and inserting searchable dimensions on top of human-designed networks known to perform well and/or careful handcrafting with human knowledge. Nonetheless, we argue that such practices inherently limit the utility of NAS. For example, in new production use cases or if the task involves new architecture paradigms, prior knowledge of effective search spaces may not be available, and such information may be expensive. More fundamentally, the current reliance on handcrafted search spaces inhibits NAS from achieving its goal of maximizing model performance "in an automated way with minimal human intervention" (Ren et al., 2021) because human expertise is still required for selecting the search space.

In this paper, we tackle this issue by proposing a simple yet effective modification to the state-of-the-art supernet training methods, where a policy that defines the search space is learned jointly with supernet weights in an end-to-end, differentiable manner during supernet training. Starting with completely uninformative priors about the search space at initialization, the policies jointly learn a subspace where NAS methods perform well within the original broad search space[1]. Our proposed approach incurs negligible additional computational overhead, requires no further retraining or fine-tuning, unlike several previous works (Pham et al., 2018), and is orthogonal with respect to most existing improvements in the supernet training pipeline

---

[1]While we largely focus on search space *shrinking* as we primarily aim to address the problem of supernet-based NAS struggling in large search spaces without prior, our method is also capable of search space *expansion*. We refer the readers to §3 for details.

(and thus offers complementary benefits). We show that our approach consistently discovers reasonable search space boundaries in huge, realistic search spaces where even the current state-of-the-art methods fail and yields high-performing supernets. Fundamentally, we argue that our method advances the applicability of NAS in search spaces beyond those commonly benchmarked in academic settings (often with extensive manual engineering) and thus represents a step towards full automation of the network architecture design pipeline.

## 2 Preliminaries

**Problem setup.** We consider a generic, typically huge NAS search space $\mathcal{A}$ that can be represented as the Cartesian product of multiple *search dimensions* $s^{(i)}$: $\mathcal{A} = \prod_{i=1}^{D} s^{(i)}$, where $D$ is the total number of search dimensions. Each search dimension $s^{(i)}$ is an ordinal variable that is chosen from a list of possible values $\{o_1^{(i)}, ..., o_{d_i}^{(i)}\}$ in an ascending order (where $d_i$ is the number of candidate choices of the $i$-th search dimension), which in turn determines a characteristic of the resulting candidate network, such as the channel depth, width, kernel size, etc. of a layer of the network, and an architecture $a \in \mathcal{A}$ can thus be represented by a $D$-dimensional vector that concatenates the search dimensions. The objective of NAS can then be formulated as a multi-objective optimization problem. For simplicity but without loss of generality, we denote the problem as a two-objective optimization problem where we aim to minimize both the validation loss $\mathcal{L}_{\text{val}}(a, W^*)$ and some cost metric ($g(a)$), such as the number of floating point operations (FLOPs) of some architecture $a$:

$$
\begin{aligned}
\min_{a \in \mathcal{A}} &\Big( \mathcal{L}_{\text{val}}(a, W^*), g(a) \Big); \\
\text{s.t.} W^* &= \arg\min_{W} \mathcal{L}_{\text{train}}(a, W), \\
g_L &\leq g(a) \leq g_U,
\end{aligned}
\tag{1}
$$

where $W$ denotes the network weights that are trained on some training set and $[g_L, g_U]$ denote some lower and upper bounds on the cost metrics that are known a-priori (for example, for deployments of neural networks on mobile devices, we typically roughly know the upper and lower bounds in terms of the number of parameters or FLOPs of the architectures we are interested in searching).

Given the multi-objective nature of the problem, we typically search for a Pareto-optimal *set* of architectures $A^* = \{a_1^*, ..., a_{|A|}^*\}$: we say that an architecture $a$ dominates another architecture $a'$ (denoted $\boldsymbol{f}(a') \prec \boldsymbol{f}(a)$) if $\mathcal{L}_{\text{val}}(a, W^*) \leq \mathcal{L}_{\text{val}}(a', W^*)$ *and* $g(a) \leq g(a')$ and either $\mathcal{L}_{\text{val}}(a, W^*) < \mathcal{L}_{\text{val}}(a', W^*)$ or $g(a) < g(a')$. Denoting $\boldsymbol{f}(a) := [\mathcal{L}_{\text{val}}(a, W^*), g(a)]^\top$, the set of Pareto-optimal architectures $A^*$ are those that are mutually non-dominated: $A^* = \{a_i^* \in \mathcal{A} \mid \nexists\, a' \in \mathcal{A} \text{ s.t. } \boldsymbol{f}(a') \prec \boldsymbol{f}(a_i^*)\}$. The Pareto front $\mathcal{P}^*$ is the image of the Pareto set of architectures: $\mathcal{P}^* = \{\boldsymbol{f}(a) \mid a \in A^*\}$.

**Supernet training with in-place KD and sandwich sampling.** As discussed, the current state-of-the-art NAS methods often rely on the ability to train supernets effectively. For a search space $\mathcal{A}$, the supernet is the largest possible sub-network $a_{\max} = [o_{d_1}^{(1)}, ..., o_{d_D}^{(D)}]$ that selects the largest candidate $o_{d_i}^{(i)}$ along all search dimensions. Letting $a_{\max}$ be parameterized by weights $W$, the goal of supernet training is that all sub-networks $a \in \mathcal{A}$ are optimized simultaneously to achieve good performance in downstream tasks. Recent works show that in-place knowledge distillation and sandwich sampling (illustrated and explained in Fig. 1) have significantly improved the supernet performance and eliminated the need for retraining (Yu & Huang, 2019; Yu et al., 2020). Formally, at time step $t$, the supernet is updated with $W_t \leftarrow W_{t-1} - \eta \nabla_W \mathcal{L}(W_{t-1})$ where $\eta$ is the learning rate and $\nabla_W \mathcal{L}(W_{t-1})$ is the gradient given by:

$$
\nabla_W \mathcal{L}(W_{t-1}) = \nabla_W \Bigg( \mathcal{L}_D([\bar{a}, W]) + \mathcal{L}_{\text{KD}}([\bar{a}, \underline{a}, W]) + \gamma \mathbb{E}_{a \sim \text{Unif}(a_1, ..., a_D)} \mathcal{L}_{\text{KD}}([\bar{a}, a, W]) \Bigg) \Bigg|_{W = W_{t-1}}
\tag{2}
$$

where the first term denotes the standard cross-entropy (CE) loss ($\mathcal{L}_D(\cdot)$) for the max network, the second term denotes the knowledge distillation (KD) loss, given by the KL divergence between the logits the max network $\bar{a}$ (the teacher) and the min network $\underline{a}$ (the student), and the final term denotes the KD loss (with

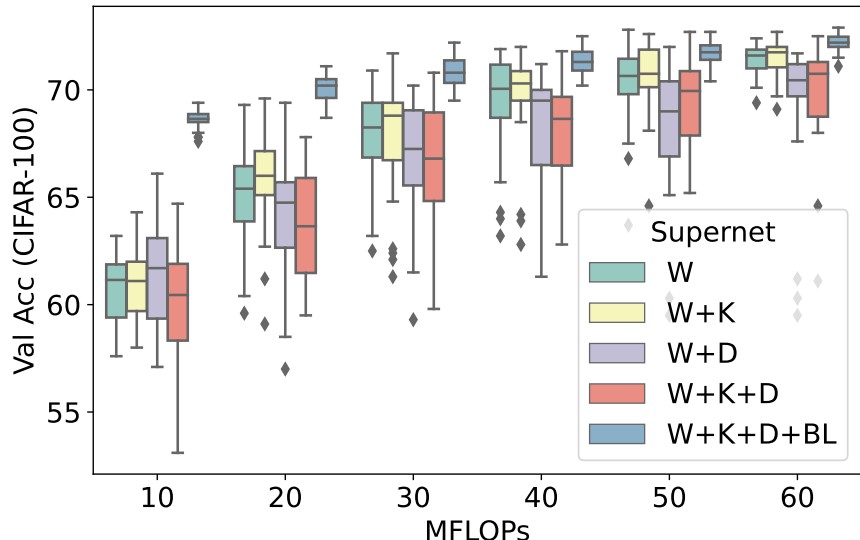

Figure 2: *Naïvely training supernets in large search spaces is ineffectual, but boundary learning closes the gap.* Distribution of CIFAR-100 validation accuracy across varying network sizes (measured in million-FLOPs (MFLOPS) in increments of 10 MFLOPs. Each box shows the distribution over 30 randomly sampled sub-networks over that FLOPs range, using the conditional sampling technique introduced in Wang et al. (2021b) from supernets trained with sandwich sampling and in-place KD in search spaces of varying sizes and complexity: `W`: searching widths (output width and expansion ratio in Inverted Residual blocks) only; `W+K`: searching widths and kernel sizes ($3\times3$, $5\times5$); `W+D`: searching widths and depth of each inverted residual block; `W+K+D`: searching widths, kernel sizes, and depths simultaneously; `W+K+D+BL`: searching all three elements, but with dynamic boundaries learning as described in §3. Boxes show medians and interquartile ranges.

the same teacher $\bar{a}$) of the random networks and $\gamma$ is a weighting factor. Unless stated otherwise, we follow previous works (Yu et al., 2020; Wang et al., 2021b;a) and always sample 2 random networks in each iteration, thus $\gamma = 2$.

## 3    Methods

**Limitations of current methods and key intuition of our proposed method.** Unlike previous techniques (Liu et al., 2019b; Pham et al., 2018; Cai et al., 2019; 2020), the key advantage of the technique described in §2 is that the resulting supernets may be directly deployed without retraining from scratch or expensive fine-tuning (one only has to re-calibrate the batch normalization statistics of the desired subnetworks without gradient back-propagation). Nonetheless, we find that the quality of the supernet training from the aforementioned technique still heavily depends on the search space design, consistent with earlier literature investigating methods on alternative search spaces (Ci et al., 2021). In Fig. 2, we train supernets consisting of MobileNet-like inverted residual blocks on search spaces of varying search space complexities. The smallest search space only contains the width search dimensions and sets other architecture parameters to fixed values that are known to perform well a priori (e.g., values in the original MobileNetv2 network specification – see Appendix A.1 for details), but the largest search space contains widths, depths, and kernel sizes simultaneously. Even though larger search spaces are more expressive and theoretically contain solutions not worse than smaller subspaces, we the empirical difficulties of supernet-based NAS methods in navigating them effectively have nevertheless been observed: as the search space becomes larger, the supernet performance generally deteriorates as measured by the accuracy of the sampled sub-networks of varying FLOP ranges. Intuitively, as the search space becomes larger and more complicated, the subnetworks become increasingly different from each other, and it is consequently more difficult for one-shot supernet

to simultaneously improve all subnetworks as the optimization directions might conflict with each other (known as *gradient conflict* (Peng et al., 2021; Gong et al., 2022)). Moreover, in a complex search space, the ground-truth smallest network might be unreasonably small or contain many harmful operators, but in naïve sandwich sampling, the supernet is still forced to sample it at each iteration even though its gradient directions might not be useful for the other subnetworks. While previous works typically bypass this problem by manually fine-tuning the search space such that the min network is still a reasonable performance lower bound, as we have discussed in §1, this is not always feasible. In the following sections, we propose a method to dynamically *learn* such boundaries without relying on expert knowledge: At a high level, the key insight lies in the fact that the boundaries of the search space are defined by the *largest* and the *smallest* sub-networks. While previous works have largely fixed their configurations a-priori, we propose to jointly optimize them as supernet training proceeds so that supernet weights can be optimized *while* promising subspaces with lesser degrees of aforementioned conflict are discovered on the fly. However, given that the sub-networks are typically specified in a discrete way, they may not be directly optimized with gradients, and we instead propose to optimize their continuous reparameterizations. We describe the mechanism of our proposed method in detail below.

**Dynamic boundaries during supernet training.** As discussed in §2, at each minibatch of supernet training, the existing supernet training method always samples the max network (i.e., the supernet) $\bar{a}$ and the smallest network $\underline{a}$ and samples two random networks $a \in \mathcal{A}$ with uniform probability. We consider a probabilistic formulation of sandwich sampling through a set of policies over the 3 different types of architectures, assuming the candidate values along each search dimension $\{o_1^{(i)}, ..., o_{d_i}^{(i)}\}$ are sorted in ascending order:

$$\text{Max: } \bar{a} \sim \prod_{i=1}^{D} \text{Cat}(\bar{a}^{(i)}|\phi^{(i)});$$

$$\text{Min: } \underline{a} \sim \prod_{i=1}^{D} \text{Cat}(\underline{a}^{(i)}|\theta^{(i)}); \tag{3}$$

$$\text{Random: } a \sim \prod_{i=1}^{D} \text{Cat}\left(a^{(i)}\middle|\omega^{(i)}\right),$$

where $\phi^{(i)}, \theta^{(i)}, \omega^{(i)} \in \Delta^{d_i-1}$, the $(d_i - 1)$-simplex, and $\text{Cat}(a^{(i)}|\cdot)$ denotes that variable $a^{(i)}$ follows a categorical distribution parameterized by $(\cdot)$. It is worth noting that here, we assume the search space consists of a Cartesian product of categorical variables with modest cardinality as this setup is by far the most common one in NAS – if, for example, one or more search dimensions are continuous or well-approximated as continuous, it may be possible to directly differentiate against the search boundary during supernet optimization. We do not, however, consider such a setup in the present paper. The baseline strategy (standard sandwich sampling) can be expressed through this formulation by setting $\text{Cat}(\bar{a}^{(i)}|\phi^{(i)})$ and $\text{Cat}(\underline{a}^{(i)}|\theta^{(i)})$ to be point-mass distributions on the largest and smallest values for each component, respectively, and setting $\omega^{(i)} = [\frac{1}{d_i}, ..., \frac{1}{d_i}]$.

Instead of deterministically selecting the smallest architecture $a_{\min} = [o_1^{(1)}, ..., o_1^{(D)}]$ and the largest architecture $a_{\max} = [o_{d_1}^{(1)}, ..., o_{d_D}^{(D)}]$ over the entire search space as the minimum and maximum points of sandwich sampling, we propose instead to *learn* the optimal boundaries. Specifically, we instantiate two policies for the min and max networks, respectively, and learn parameters $\theta, \phi \in \mathbb{R}^{\sum_{i=1}^{D} d_i}$ controlling the min and max bounds of sandwich sampling, respectively. Although it is possible to use a more expressive policy to handle the dependencies between different search dimensions, empirically, we find using the independent policies outlined above Eq. (3) performs well. We propose to optimize the *policy* jointly weights $\{\theta, \phi\}$ alongside supernet weights $W$ by first modifying losses ($\mathcal{L}_D$ and $\mathcal{L}_{\text{KD}}([\underline{a}, W])$) associated with the max and min networks,

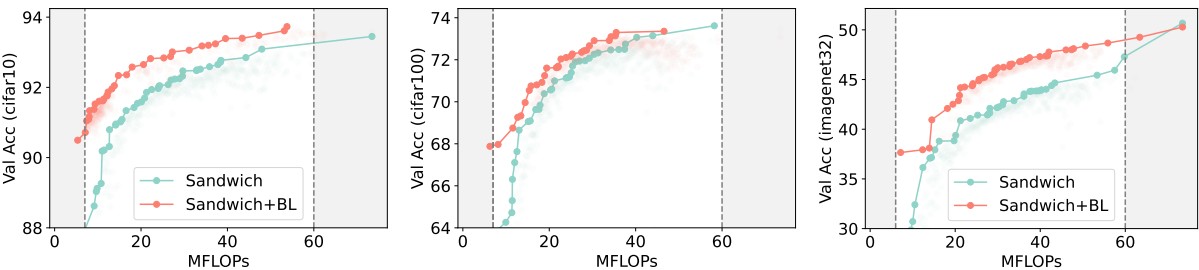

Figure 3: Comparison of the Pareto fronts of top-1 accuracy vs. million FLOPs (MFLOPs) of architectures in the width-only search space (§4.1) searched with standard techniques described in §2 with (`Sandwich + BL`) and without (`Sandwich`) our proposed boundary learning in, from left to right, CIFAR-10, CIFAR-100, and ImageNet-D. The upper and lower bounds in terms of MFLOPs ($\{g_L, g_U\}$ in Eq. (1)) are set to $[6, 60]$ MFLOPs in all experiments (the MFLOPs range of interest is marked in white in the figures).

respectively, with:

$$\begin{aligned} \text{Max: } & \mathbb{E}_{\bar{a}\sim\bar{\pi}(\bar{a}|\phi)}\Big[\mathcal{L}_D([\bar{a}, W]) + \lambda\ell\big(g_U - g(\bar{a})\big)\Big]; \\ \text{Min: } & \mathbb{E}_{\underline{a}\sim\underline{\pi}(\underline{a}|\theta)}\Big[\mathcal{L}_{\text{KD}}([\bar{a}, \underline{a}, W]) + \lambda\ell\big((g(\underline{a}) - g_L)\big)\Big], \end{aligned} \tag{4}$$

where $\ell(y) = \max(0, y)$ is the hinge loss to penalize the max and min networks for moving too far away from the pre-determined range defined in terms of some cost metric (FLOPs in this paper). These penalties are in place to avoid missing out on portions of the Pareto front in the region of interest, and $\lambda$ controls the strength of penalization (set to 5 throughout). It is worth noting that the min and max networks are now both sampled from some parameterized policies. Thus, the samples (and parameterized distributions) may differ at each gradient update step.

We retain the uniform sampling strategy for the regular random architectures, but condition on the learned min and max policies. Formally, at each search dimension $i \in [1, D]$ we first, compute the cumulative density functions (CDFs) of both the min and max policies. For the $j$-th choice of the $i$-th dimension, the CDFs of the max and min policies are simply:

$$\begin{aligned} \text{Max CDF: } & F(\phi^{(i)})_j = \sum_{k=1}^{j} \phi_k^{(i)}; \\ \text{Min CDF: } & F(\theta^{(i)})_j = \sum_{k=1}^{j} \theta_k^{(i)} \ \forall \ j \in [1, d_i], \end{aligned} \tag{5}$$

where $\phi^{(i)} = [\phi_1^{(i)}, ..., \phi_{d_i}^{(i)}]$ and $\theta^{(i)} = [\theta_1^{(i)}, ..., \theta_{d_i}^{(i)}]$. The parameters $\omega = [\omega^{(1)}, ..., \omega^{(D)}]$ for the random policy random architectures are set to:

$$\text{Random: } \omega_j^{(i)} := c^{(i)} \mathbb{1}(1 - F(\phi^{(i)})_j - \tau) \cdot \mathbb{1}(F(\theta^{(i)})_j - \tau), \tag{6}$$

where $\mathbb{1}(\cdot)$ is the Heaviside step function, $\tau$ is some threshold (set to 0.5 in all our experiments), and $c^{(i)}$ is a normalization factor $c^{(i)} = \frac{1}{\sum_j \omega_j^{(i)}}$. This strategy essentially performs uniform sampling but only in the regions in the search space bounded by the min and max architectures with high probabilities. Specifically, $\omega_j^{(i)}$ is 1 if and only if the cumulative probability that the index of max network for dimension $i$ is greater than $j$ *and* the cumulative probability that the index of min network for dimension $i$ is less than $j$ are *both* greater than $\tau$.

**Policy learning.** To reflect an uninformative prior on the search space, we initialize the policy parameters to imitate the baseline sandwich sampling strategy by assigning the most probability to each search

dimension's smallest/largest choices for the min/max policies, respectively. Formally, for the $i$-th search dimension, we initialize the policy weights as follows:

$$\text{Max: } \phi^{(i)} \leftarrow \sigma\left(\left[a_0, a_0 + \epsilon_0, ..., a_0 + \sum_{j=1}^{d_i-1} \epsilon_0\right]^\top \Big/ T\right);$$

$$\text{Min: } \theta^{(i)} \leftarrow \sigma\left(\left[a_0 + \sum_{j=1}^{d_i-1} \epsilon_0, a_0 + \sum_{j=1}^{d_i-2} \epsilon_0, ..., a_0)\right]^\top \Big/ T\right), \tag{7}$$

where $a_0$ is a constant set to 0.1, $\epsilon_0$ are positive random weights sampled from a normal distribution $\mathcal{N}(0, 10^{-3})$, $\sigma(\cdot)$ denotes the softmax function, and $T$ is the softmax temperature (we use a $T = 0.1$ throughout). This initialization formula ensures that the weights sum up to 1 and that at initialization, the probability masses that we assign on each candidate value are in ascending order (with the largest probability for the largest possible choice) for the max policy and descending order for the min policy along each search dimension. It is worth noting that the above initialization recipe can be adapted to incorporate prior knowledge, if available, by simply re-allocating the probability masses appropriately. An exemplary use case of this is if one would like to adaptively *expand* search spaces in addition to shrinking: one may define an even larger search space and initialize the most probability mass of the max policy on a candidate value other than the largest choice; this allows the policy to potentially grow the search space as joint supernet and learning space boundary learning takes place.

To enable learning $\{\phi, \theta\}$ and $W$ jointly in a differentiable manner using Eq. (4), we need to compute gradients $\nabla_\phi \mathbb{E}_{\bar{a} \sim \bar{\pi}(\bar{a}|\phi)}\left[\mathcal{L}_D([\bar{a}, W]) + \lambda\ell(g_U - g(\bar{a}))\right]$ and $\nabla_\theta \mathbb{E}_{\underline{a} \sim \underline{\pi}(\underline{a}|\theta)}\left[\mathcal{L}_{\text{KD}}([\underline{a}, W]) + \lambda\ell((g(\underline{a}) - g_L))\right]$. It is worth noting that the partial derivatives for the activated weights (i.e., the weights included in the current subnetwork) are the same as the partial derivatives in Eq. (2), and the partial derivatives for the inactivated weights are zero since the modification to the loss function is independent of $W$. To achieve this, we use the *Gumbel-softmax* relaxation (Jang et al., 2017) for the categorical policies in Eq. (3) for gradient approximation. Specifically, we use the straight-through variant where we always discretize an architecture during a forward pass (i.e., sample exactly from the categorical distributions) but backpropagate using the gradient of the non-one-hot Gumbel-softmax sample. Although other gradient estimators such as the score-based (Williams, 1992; Fu, 2006) and measure-valued (Mohamed et al., 2020) alternatives may also be used, we opt for Gumbel-softmax because we have a differentiable objective function and pathwise methods are likely to yield lower gradient variances (Carvalho et al., 2021). We also use an analytical formula to compute FLOPs during the forward pass to ensure the constraint term in Eq. (4) is also differentiable. In the case where no such analytical formula is available for an alternative cost metric, such as latency in mobile devices, we may still use techniques such as pre-computed look-up tables proposed in FBNet (Wu et al., 2019) or differentiable latency modules (Xu et al., 2020) to retain differentiability. The overall pipeline of the proposed algorithm is presented in Algorithm 1, with the key differences from the standard sandwich sampling highlighted in magenta.

---

**Algorithm 1** Sandwich sampling with boundary learning

---

Initialize policy weights of $\{\phi, \theta\}$ according to Eq. (7).
**while** not converged **do**
  Sample a mini-batch of data $\mathcal{B}$ from train data $\mathcal{D}$.
  Draw a max network $\bar{a} \sim \bar{\pi}(\bar{a}|\phi)$.
  Train the max network $\bar{a}$ with true labels from $\mathcal{B}$
  Draw a min network $\underline{a} \sim \underline{\pi}(\underline{a}|\theta)$ to mimic the max network with the KD loss
  Compute & store gradient of policy parameters $\phi$ and $\theta$ w.r.t. loss in Eq. (4)
  Sample $\max(n - 2, 0)$ random network(s) according to Eq. (3) with a training objective to mimic the output logits of the max network with the KD loss.
  Update supernet weights $W$ and policy parameters $\phi, \theta$ via gradient descent.
**end while**

---

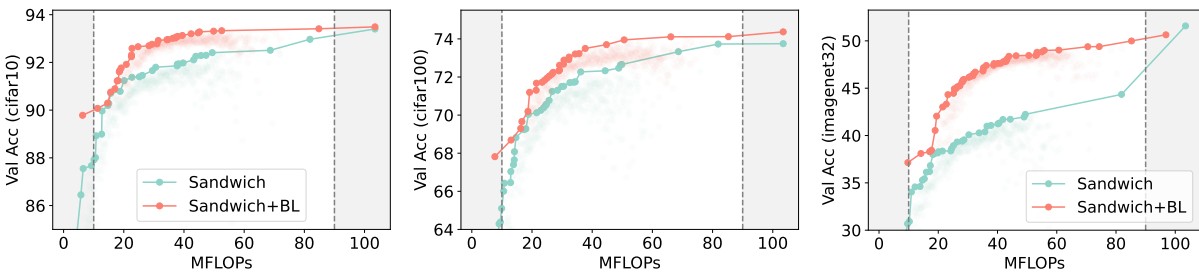

Figure 4: Comparison of the Pareto fronts of top-1 accuracy vs. MFLOPs of architectures for Weight-sharing NAS experiments (§4.2) in CIFAR-10, CIFAR-100, ImageNet-D, and the original ImageNet (left to right). The upper and lower bounds in terms of MFLOPs for all datasets are set to $[10, 90]$.

**Analysis of computing costs.** Our method incurs minimal additional computational overhead over standard supernet training: The only additional cost is the cost to train & backpropagate the gradients of the policies (i.e., $\phi$ and $\theta$ in Eq. 3; note that $\omega$ are not free parameters and are linked to $\phi$ and $\theta$ through Eq. 6). There are a total of $2\sum_{i=1}^{D} d_i$ such free parameters parameterizing the two categorical distributions for the max and min policies, where $D$, the number of search dimensions in the search space, is typically $\leq 50$, and $d_i$, the number of choices of the $i$-th search dimension, is typically $\leq 10$. The additional number of parameters to train is thus $\sim \mathcal{O}(10^3)$, which is negligible compared to the number of parameters of modern neural networks, which is at least $\mathcal{O}(10^6)$.

## 4 Experiments

In this section, we i) empirically investigate the effectiveness of boundary learning in various search spaces inspired and generalized from well-known architectures like Slimmable networks (Yu et al., 2019) and MobileNet-family CNNs and ii) analyze the learned boundaries. It is worth noting that we deliberately focus on expanded, less hand-tuned search spaces rather than the existing, commonly used search spaces. This is because, as discussed in §1, existing search spaces, such as those used in previous works like Wang et al. (2021b) and Wang et al. (2021a), feature per-layer handcrafted boundaries and some fixed search dimensions. These search spaces are often heavily engineered by human experts *such that* NAS works well, and it is unknown how well the results may generalize. Thus, rather than focusing on search spaces that have undergone extensive handcrafting, we focus on less hand-tuned search spaces as a closer proxy to *arbitrary*, novel search spaces likely encountered in practical settings. We also opt not to study cell-based search spaces such as the DARTS (Liu et al., 2019b) or the NAS-Bench spaces (Ying et al., 2019; Dong et al., 2021) because the objective of our work is to improve the retraining-free NAS and to identify *a family of Pareto-optimal architectures over a wide range of costs* similar to previous works like Wang et al. (2021b), Yu et al. (2020) and Cai et al. (2020). In contrast, the aforementioned cell-based search spaces typically feature a small variation in model size, and usually, the objective is to identify a single best architecture.

### 4.1 Width-only Search Space

**Search space.** We first experiment on a width-only search space that we adapt and enlarge from the one introduced in Slimmable networks (Yu et al., 2019). The search space admits a wide range of channel width configurations with other search dimensions, such as kernel sizes and depths, fixed. In contrast with Slimmable networks, we follow Chin et al. (2021) and allow the widths of the different layers to be different: instead of forcing all layers to have the same width (measured as the fraction of the width of activated neurons to the maximum width of each layer). The original search space essentially couples all the sub-networks to a manually specified supernet configuration, which implies a rather strong prior on the search space. Our relaxed search space significantly weakens that coupling and yields a much more complex search space over more diverse architectures, as the width of each layer width becomes a free search dimension, and the number of possible subnetworks scales combinatorially.

**Settings.** Our search space is largely adapted from the MobileNetv2 variant of the Slimmable networks search space, and we search for the output width and the width of the expanded intermediate convolution layers for all stages as well as the initial convolution channels. As in Yu & Huang (2019), we set the maximum width $w_{\max}^{(i)}$ of each layer and allow the width to be chosen from $w^{(i)} = \{0.125, 0.25, ..., 0.875, 1\} \times w_{\max}^{(i)}$ along each search dimension. Note that this is a significantly larger range than the search space originally proposed (Yu et al., 2019), and the corresponding search space contains many more small networks not found in the original search spaces, which are useful for additional devices with modest computational powers. Apart from the restriction that the output channel width of $i$-th layer must match the input width of the $(i+1)$-th, we place no further constraints on what width each layer may take. We train all models for 120 epochs using SGD for the supernet weights $W$ and Adam (Kingma & Ba, 2015) for the policy weights ($\phi$ and $\theta$), and we use a single set of hyperparameters for all our experiments without further task- or model-specific hyperparameter tuning (see Appendix A.2 and A.3 for the implementation details). After the supernet training, we closely follow Wang et al. (2021a) and Wang et al. (2021b) to run an adapted version of genetic algorithm *within the identified subspace defined by the learned boundaries* to identify the set of non-dominated Pareto-optimal architectures that trade between Top-1 accuracy and the number of FLOPs (although a more sample efficient method, such Bayesian optimization methods adapted to discrete search spaces and/or multiobjective optimization settings (Ru et al., 2021; Daulton et al., 2021; 2022), may be used instead). For the width-only search space, we experiment on CIFAR-10, CIFAR-100 (Krizhevsky, 2009), and a downsampled variant of ImageNet (denoted as ImageNet-D) (Chrabaszcz et al., 2017), which is the full ImageNet dataset but only with resolution downsampled from $224 \times 224$ to $32 \times 32$, thereby making it an even more challenging task than the full-resolution ImageNet.

**Results.** We summarize the results in Fig. 3. On all datasets, we find that boundary learning yields significant improvements of the Pareto front over vanilla supernet training with the recipe described in §2.

## 4.2 Weight-sharing NAS

**MobileNetv2-like search space.** Going beyond width-only search spaces, we further investigate MobileNetv2-like search spaces that are more consistent with modern weight-sharing NAS methods that further incorporate the depth and kernel size dimensions (Wang et al., 2021b;a). In addition to the width dimensions in §4.1, we also incorporate dynamic kernel sizes ($3 \times 3$ and $5 \times 5$) and dynamic depth per stage, and the detailed specification may be found in Table 1. To simulate real-life NAS search spaces with scarce prior knowledge, we design a search space that is orders of magnitude larger and more complex than common search spaces featured in the literature and features significantly less handcrafting. Specifically, unlike existing works that often only allow widths and depths to vary in narrow, carefully selected, and often layer-specific ranges, we allow width configurations to vary widely, and we adopt the same range for the possible depth for each stage. As a result, our search space does not rely on informed priors from human trial-and-error, and thus is far more generalizable than previous bespoke search spaces defined using prior knowledge which, as some previous works have shown (Tu et al., 2022), is likely task-specific and might not necessarily transfer into novel tasks.

We report the results in the search space described in Table 1 in Fig. 4 where boundary learning has significantly improved the supernet across FLOPs ranges. It is also noteworthy that the margin of gain in ImageNet-D is particularly large, possibly due to the challenging nature of the task, which warrants a careful search space design. We additionally show the progression of the Pareto fronts recovered as the second-stage genetic algorithm proceeds in Fig. 12 in App. B, which demonstrates an *anytime* improvement of BL over baseline during search.

**MobileNetv3-like search space.** We also conduct preliminary experiments on an even larger and more challenging search space inspired by MobileNetv3 (Howard et al., 2019): in addition to the search dimensions in Table 1, we also include the search for the activation function ({ReLU, swish}) and whether to use squeeze-and-excite (SE) module (Hu et al., 2018) for Block 1-6, thereby adding a further 12 dimensions to the search space – this is in contrast to the search spaces featured in previous works like Wang et al. (2021a) which *hard-code* these design decisions manually. We show the results in Fig. 5, where we find the margin of

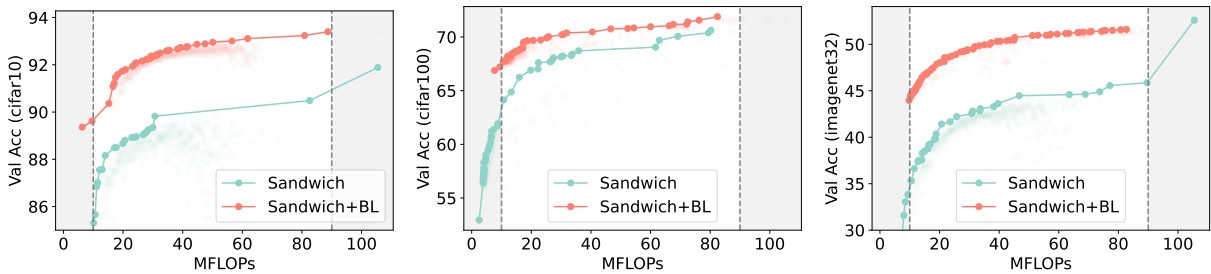

Figure 5: Comparison of the Pareto fronts of top-1 accuracy vs. MFLOPs of architectures in the MobileNetv3-like search spaces in CIFAR-10, CIFAR-100, and ImageNet-D.

improvement to be even larger in this search space: even though the elements introduced in MobileNetv3, such as the SE module, have been shown to improve the performance significantly, the additional complexity has a crippling effect on the naïve supernet training, causing it to perform even worse compared the results reported in Fig. 3 and 4. On the other hand, boundary learning largely restored the supernet performance, demonstrating its efficacy despite increased complexity.

Table 1: Specification of the search space used in weight-sharing NAS experiments in §4.2 *Intermediate* refers to the possible widths of the expanded convolution module in inverted residual blocks. *#SD* denotes the number of search dimensions of a stage.

| Stage | Intermediate | Output | Depth | Kernel | #SD |
|---|---|---|---|---|---|
| Head | - | 4-32 | - | 3, 5 | 2 |
| Block1 | 12-96 | 2-16 | 1 | 3, 5 | 3 |
| Block2 | 24-144 | 3-24 | 1-4 | 3, 5 | 7 |
| Block3 | 24-144 | 3-24 | 1-4 | 3, 5 | 7 |
| Block4 | 48-384 | 8-64 | 1-4 | 3, 5 | 7 |
| Block5 | 72-576 | 12-96 | 1-4 | 3, 5 | 7 |
| Block6 | 120-960 | 20-160 | 1-4 | 3, 5 | 7 |
| Block7 | 240-1920 | 40-320 | 1 | 3, 5 | 3 |

## 4.3 Analysis of Learned Subspaces

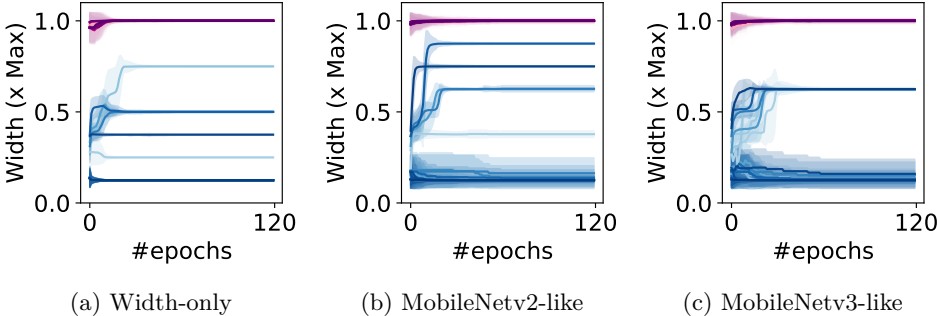

(a) Width-only      (b) MobileNetv2-like      (c) MobileNetv3-like

Figure 6: Evolution of the **widths** (expressed as a fraction of the max width of each layer) for the Max and Min networks as a function of training epochs in search spaces discussed in §4.1 and §4.2. Shades denote one standard deviation. The strength of the color denotes the depth of the search dimension (deeper search dimensions have darker colors).

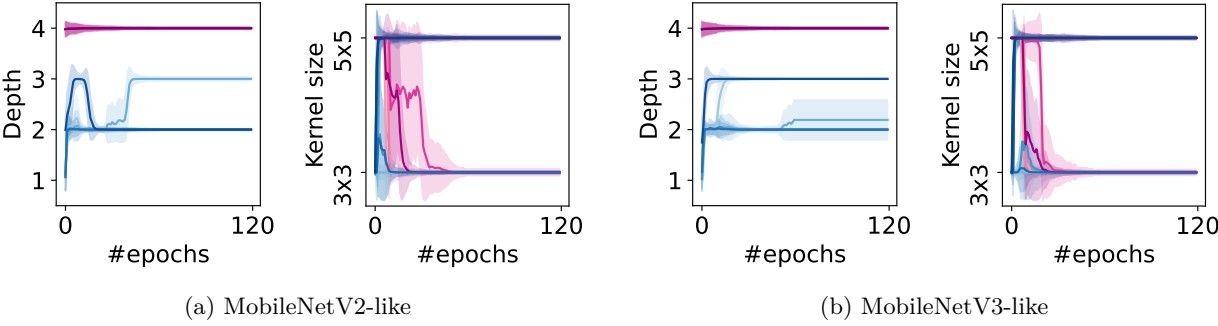

(a) MobileNetV2-like             (b) MobileNetV3-like

Figure 7: Evolution of the **depths** and **kernel sizes** of expected Max and Min networks in MobileNetV2-like and MobileNetV3-like search spaces described in §4.2.

To understand the effectiveness of boundary learning, we also analyze the boundaries learned and how they evolve as the training progresses. Given the probabilistic nature of the policies, we compute the trajectories of the *expected* max and min architectures by marginalizing the policy probabilities along each search dimension alongside their standard deviation in Fig. 6 – 8 where we analyze the ImegeNet-D task. The reader is referred to Appendix B for additional visualizations and analyses of more tasks and datasets.

For the max architectures, we observe generally on width and depth dimensions (Fig. 6 and 7a) that the expected max architecture remains at the largest possible values as the training progresses. We also observe that the polices become increasingly confident over time, as shown by the decreasing standard deviation of the sampled architectures over epochs. It is intuitive that the ground-truth, deepest, and widest network may be the best max network regardless of what FLOPs range that we ultimately are interested in since it acts as a teacher. This, however, is not necessarily the case when additional search dimensions are included in the search space described in §4.2. Intuitively, while we expect a wider and/or deeper network to have a larger learning capacity and hence to be a better teacher, it is

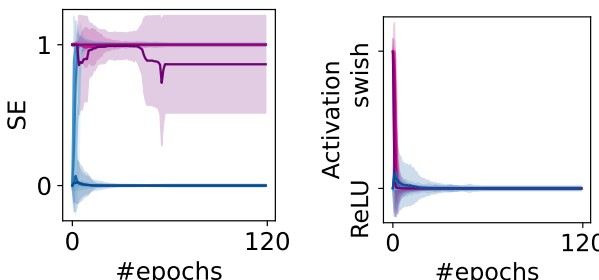

Figure 8: Evolution of the **SE module presence** (0 denotes absence of SE module for a stage and 1 denotes presence) and **activation function** choice for the Max and Min networks in the MobileNetV3-like space described in §4.2.

less obvious, for example, whether a network with larger kernel sizes will categorically perform better than an alternative architecture with smaller kernel sizes. Similarly, in the MobileNetv3 search space, there is no obvious a-priori reason to prefer an activation function over another. Indeed, we observe that from Fig. 7b and 8. In these cases, the policy has learned a max architecture that is different from the ground-truth maximum by adopting smaller, 3×3 kernel sizes at several initial stages. Interestingly, this corroborates with the design of MobileNetv3, which also uses smaller kernel sizes at the shallower layers of the networks and larger ones towards the end.

The min network can be seen as the performance lower bound, and we expect it to have a larger influence on the overall supernet training. Indeed, we observe that in all cases, the policy has quickly learned to enlarge the min network away from the smallest possible architecture in the search space, which is often unreasonably small. Furthermore, we find that the policy has often learned a highly non-uniform lower bound. While in some search dimensions, the learned min does not move from the initial values at all, in other search dimensions, it shifts more aggressively. We find that the dimensions where the learned does change often coincide with the more important dimensions from our domain knowledge. For example, in search spaces where depths are included as search dimensions (Fig. 7a), the policy always opts to increase depths first almost across all stages. This is reasonable, given the empirical findings suggesting that depths are likely the most important search dimensions both in NAS (e.g., Shu et al. (2020) show that a narrower but

deeper network generalizes better than a wider but shallower network of similar sizes in cell-based NAS). Our analysis suggests that the policies have learned meaningful boundaries, often discovering patterns consistent with domain knowledge but independently without prior information. This is crucial, as it validates the potential of boundary learning in arbitrary, potentially unknown search spaces that we might be interested in.

### 4.4 Ablation Studies

We also conduct ablation experiments by comparing i) learning both max and min networks vs. learning the min networks only, ii) whether to back-propagate gradients from the Random architectures to the min and max policies during training, iii) the comparison between boundary learning with baseline supernet training with a longer number of epochs , iv) whether the benefit of BL persists when an alternative optimizer other than SGD (we studied AdamW (Loshchilov & Hutter, 2018)) is used and v) sensitivity of BL w.r.t different random seeds. We find that learning max and min modestly outperforms learning min only, and back-propagating gradients from Random architectures generally worsen performance, likely due to the additional noise introduced in the gradient estimates. We also find that while doubling the number of training epochs improves supernet performance trained via the baseline protocol, it is still outperformed by boundary learning presented in this section, even though the latter is approximately half as expensive. The readers are referred to Appendix C for details.

## 5 Related Works

In this section, we give a detailed description of the related works, broadly categorized as either (i) aiming to improve the supernet training pipeline or (ii) aiming to improve the search space itself.

**Improvements on supernet training.** Most of the endeavors in NAS have been trying to improve the search methodology, and given the dominance of supernet-based methods in modern NAS, most previous works have focused on improving supernet training. In particular, since the initial proposal of retraining-free and finetuning-free supernet training strategy (Yu & Huang, 2019), various improvements have been proposed in different stages of the pipeline (and in light of the vast literature, we only discuss the most relevant works in this section): AttentiveNAS (Wang et al., 2021b) and Joslim (Chin et al., 2021) essentially propose non-uniform sampling strategies for the Random architectures during training (i.e., $a$ in Eq. (3) by focusing on the Pareto-optimal architectures discovered so far or with a Bayesian optimization agent, respectively. Several other works focusing on alternative goals have also proposed similar search space adaptation ideas: For example, Neural Architecture Transfer (NAT) and NSGANetv2 (Lu et al., 2020; 2021) focus on the fast adaptation of neural architecture for diverse tasks and/or many, potentially conflicting objectives. To address the problem of large supernet search spaces and the consequent problem of insufficient exploration, the authors proposed to focus on the promising subnetworks recommended by a predictor/evolutionary algorithm instead of uniformly sampling from all subnetworks, thereby constraining the attention on a subspace of promising sub-networks. However, similar to Wang et al. (2021a), if applied in the context of sandwich sampling, these techniques invariably aim to improve the sampling strategy of the *random* architectures (Eq. 3) network as opposed to the Min and Max architectures that we focus on. AlphaNet (Wang et al., 2021a) replaces the KL divergence in the KD loss of Eq. (2) with an adaptive alpha divergence. NASViT (Gong et al., 2022) extends the aforementioned training techniques to hybrid CNN-Vision Transformers (ViT) search spaces and addresses the issue of gradient conflict. However, our work is search space-agnostic and addresses a different aspect in the NAS pipeline, and thus is orthogonal and offers potentially combinable benefits with respect to all of the aforementioned works, as we show in a series of preliminary experiments where we both compare against and combine our method with some of the related works in Appendix B.

**Improvements on search spaces.** Following the recent discoveries of the sensitivity of NAS methods to search spaces, there has also been a line of work that aims to improve the search space itself (our work falls into this category): Earlier works (Liu et al., 2018; Perez-Rua et al., 2018; Ru et al., 2020) explore the ideas of search space evolution and/or optimization within the query-based NAS paradigm, which is typically very computationally expensive. More recently, Ci et al. (2021), Chen et al. (2021a) and Xia et al. (2022) consider

search spaces evolution of CNNs and ViTs, but the methods proposed require training of multiple supernets. Lastly, several previous works also explore search space adaptation on the fly similar to us: for example, Hu et al. (2020) use an angle-based metric; Nayman et al. (2019) use expert advice; Noy et al. (2020) use annealing; Chen et al. (2021b) gradually increase channel width. However, these methods are often heuristic- or scheduling-based and are often search space-specific, whereas our method derives signal for search space adaptation from the training loss directly along with the supernet weights. Other works, such as Zhao et al. (2021) and Su et al. (2021), alleviate the training difficulty of supernets in large search spaces by dividing the supernets into sub-supernets and training them separately. These approaches inevitably lead to a trade-off between performance and efficiency as multiple supernets now need to be trained; in contrast, our method retains the one-shot nature of supernet training and has little impact on the overall efficiency.

## 6    Conclusion

We propose a novel method that jointly learns and refines the search space boundary with supernet optimization in an end-to-end, differentiable manner by introducing learnable policy modules on top of the supernet. Despite its simplicity, we show its effectiveness in a range of tasks where our method drastically improves NAS performance in large and complicated search spaces, even though existing methods struggle. Ultimately, we hope our work opens new possibilities in finding general-purpose NAS methods that function in any search space, even without prior knowledge.

**Limitations and future work.**    We note that although our method is conceptually generic, we have only considered image classification tasks in CNN-based search space in this paper, and an immediate next step is to extend the method to other search spaces (e.g., the one in Gong et al. (2022)) and other tasks (e.g., those in NAS-Bench-360 (Tu et al., 2022)). Furthermore, in designing the methodology, we also introduce a number of hyperparameters in the pipeline, such as $T, a_0, \epsilon_0$ in Eq. (7) and $\lambda$ in Eq. (4). While we use a consistent set of these hyperparameters throughout and do not tune them, their impact on the performance and the way to potentially automate their choices remain to be investigated. We defer these to future work.

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

# A Implementation Details

Table 2: Specification of the largest search space used in the paper. *Intermediate* refers to the possible widths of the expanded convolution module in inverted residual blocks. *#SD* denotes the number of search dimensions of a stage. *SE* refers to the choice of whether to have a Squeeze-and-Excite module for that stage and *Activation* denotes the choices of the activation function.

| Stage | Intermediate | Output | Depth | Kernel | SE | Activation | #SD |
|-------|--------------|--------|-------|--------|-------------|--------------|-----|
| Head | - | 4-32 | - | 3, 5 | - | - | 2 |
| Block1 | 12-96 | 2-16 | 1 | 3, 5 | True, False | ReLU, Swish | 5 |
| Block2 | 24-144 | 3-24 | 1-4 | 3, 5 | True, False | ReLU, Swish | 9 |
| Block3 | 24-144 | 3-24 | 1-4 | 3, 5 | True, False | ReLU, Swish | 9 |
| Block4 | 48-384 | 8-64 | 1-4 | 3, 5 | True, False | ReLU, Swish | 9 |
| Block5 | 72-576 | 12-96 | 1-4 | 3, 5 | True, False | ReLU, Swish | 9 |
| Block6 | 120-960 | 20-160 | 1-4 | 3, 5 | True, False | ReLU, Swish | 9 |
| Block7 | 240-1920 | 40-320 | 1 | 3, 5 | True, False | ReLU, Swish | 5 |

## A.1 Search Spaces

The search spaces used in this paper are largely based on the MobileNet search spaces adapted from the specification listed in Table 2.

**Width-only.** The width-only search space (§4.1 and the `W` search space in Fig. 2 only have *Intermediate* and *Output* search dimensions activated, and the depth dimensions are fixed at values $\{-, 1, 2, 3, 4, 3, 3, 1\}$ from Head to Block7, respectively and kernel size is fixed at $3 \times 3$. SE is `False`, and activation is set to ReLU for all stages. For the experiments done in Fig. 2, `W+K` denotes the search space with *Intermediate, Output* and *Kernel* dimensions activated, `W+D` denotes the one with *Intermediate, Output* and *Depth* as searchable dimensions and `W+K+D` denotes the one with all three as searchable dimensions.

**Weight-sharing NAS.** The MobileNetv2 search space described in Table 1 in the main text is a subspace of 2 with SE and Activation set to `False` and ReLU, respectively. The MobileNetv3 search space has the full space specification as described in Table 2.

## A.2 Training Protocol

We train the supernets for 120 epochs for all experiments using the SGD optimizer with a Nesterov momentum of 0.9. We first use a linear learning rate warm-up schedule for the first 5 epochs, with the learning rate increased from $10^{-5}$ to $10^{-1}$, followed by a cosine annealing learning rate decay. We follow previous works (Wang et al., 2021a;b) and apply a dropout of probability of 0.2 and drop connect probability of 0.2 on the supernet for additional regularization. On CIFAR-10 and CIFAR-100 datasets, we use a weight decay of $5 \times 10^{-4}$ for the non-batch normalization (BN) weights and 0 for the BN bias, a batch size of 256, and a maximum learning rate of 0.1. On ImageNet-D, we use a weight decay of $10^{-5}$ for the non-BN weights and 0 for the BN weights, a batch size of 1024, and a maximum learning rate of 0.4. We use Adam optimizer for the policy weights with a learning rate of $5 \times 10^{-4}$, weight decay of 0, and other parameters unmodified at their default values.

## A.3 Evaluation Protocol

After the training is complete, we search for the Pareto front between accuracy and FLOPs using a genetic algorithm adapted from previous works (Yu et al., 2020; Wang et al., 2021b). Specifically, we i) randomly sample 256 sub-networks (including the max and min, which are always sampled) from the supernet and compute their FLOPs and their validation accuracy, and we select the architectures on the Pareto front for this initial set of architectures; ii) we apply crossover (given two different parent subnetworks on the Pareto

front $a_1 = [s_1^{(1)}, ..., s_1^{(D)}]$ and $a_2 = [s_2^{(1)}, ..., s_2^{(D)}]$, we build $a_3$ where each search dimension is sampled from the values in the two parent subnetworks: $s_i^{(3)} \sim \text{Unif}(s_i^{(1)}, s_i^{(2)})$) and mutation (given a Pareto architecture $a$, on each search dimension, we randomly change its value to another value with probability of 0.1). We fix the crossover and mutation sizes to 128, thus generating a new set of 256 sub-networks. We then evaluate the performance of the new sub-networks, repeat the second step for 20 epochs, and report the final Pareto front at the end of the architecture selection stage.

## B  Additional Experimental Results

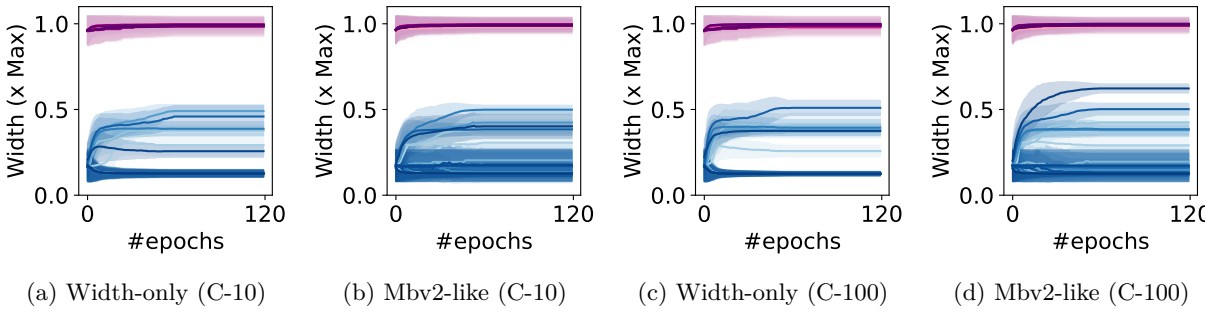

(a) Width-only (C-10)  (b) Mbv2-like (C-10)  (c) Width-only (C-100)  (d) Mbv2-like (C-100)

Figure 9: Evolution of the **widths** (expressed as a fraction of the max width of each layer) for the Max and Min networks as a function of training epochs in search spaces discussed in §4.1 and §4.2 for **CIFAR-10** and **CIFAR-100**. Shades denote one standard deviation. The strength of the color denotes the depth of the search dimension (deeper search dimensions have darker colors. Mbv2: MobileNetv2-like search space (§4.2).

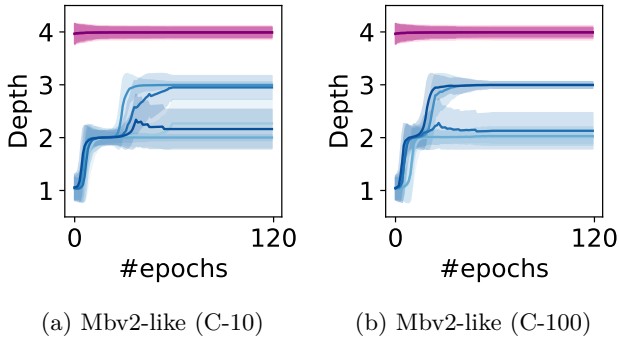

(a) Mbv2-like (C-10)  (b) Mbv2-like (C-100)

Figure 10: Evolution of the **depths** (expressed as a fraction of the max width of each layer) for the Max and Min networks as a function of training epochs in search spaces discussed in §4.1 and §4.2 for **CIFAR-100**.

**Additional visualizations.**  In this section, we report additional experimental results by presenting visualizations and analyses of the learned subspaces on more datasets (presented in Fig. 9 − 11). We observe that most of the high-level trend described in §4.3 holds, such as the propensity for the policy to quickly increase the width of the min network over time. One notable exception is that for the CIFAR datasets, the max networks for all search dimensions, including the kernel size dimension, remain at the largest possible value, unlike the ImageNet-D case where the kernel size of the max network decreases to 3×3 for some layers.

**Further analysis of Pareto fronts.**  Complementary to the figures in the main text, we present additional statistics of the Pareto fronts, measured in terms of *hypervolume* in Table 3. Specifically, we compute the

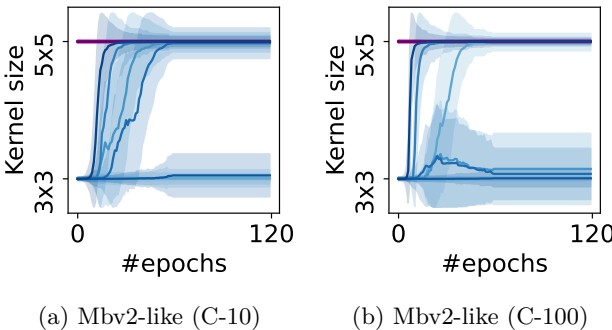

(a) Mbv2-like (C-10)    (b) Mbv2-like (C-100)

Figure 11: Evolution of the **kernel sizes** (expressed as a fraction of the max width of each layer) for the Max and Min networks as a function of training epochs in search spaces discussed in §4.1 and §4.2 for **CIFAR-100**.

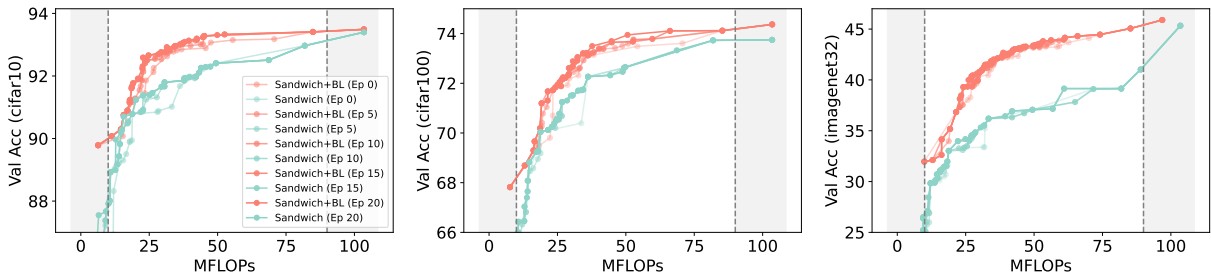

Figure 12: Comparison of the Pareto fronts recovered *over time* (at {0 (start), 5, 10, 15 and 20 (termination)}-th epochs, as signified by the different shades of the lines) of baseline sandwich sampling vs. when our proposed BL is applied on MobileNetv2 search space.

hypervolume with the *dimension sweep* algorithm from Fonseca et al. (2006). Hypervolumes are computed against a *reference point* – we automatically determine the reference point using the technique presented in Ishibuchi et al. (2011). In practice, we achieve both using the hypervolume utilities in the BoTorch (Balandat et al., 2020) package. From Table 3, we find that BL consistently improves the hypervolume of the Pareto fronts, consistent with our visual observations.

|  |  | CIFAR-10 | CIFAR-100 | ImageNet-D |
|---|---|---|---|---|
| MobileNetv2 | Ref. point | [79.0, 113.7] | [50.1, 113.7] | [12.5, 113.7] |
|  | Sandwich | 1421.60 | 2364.66 | 2688.70 |
|  | Sandwich+BL | **1461.49** | **2437.56** | **3119.92** |
| MobileNetv3 | Ref. point | [76.64, 115.8] | [51.1, 90.5] | [12.5, 113.7] |
|  | Sandwich | 1429.48 | 1471.02 | 2688.70 |
|  | Sandwich+BL | **1749.16** | **1606.12** | **3119.92** |

Table 3: Hypervolume of the Pareto fronts (higher is better) obtained by Sandwich and Sandwich + BL w.r.t the reference point (presented as 2-tuples in the format of [Accuracy, MFLOPs].

**Progression of Pareto fronts.** We show the progression of the Pareto fronts recovered at the {0, 5, 10, 15, 20}-th search epoch of both sandwich sampling with and without the proposed boundary learning in

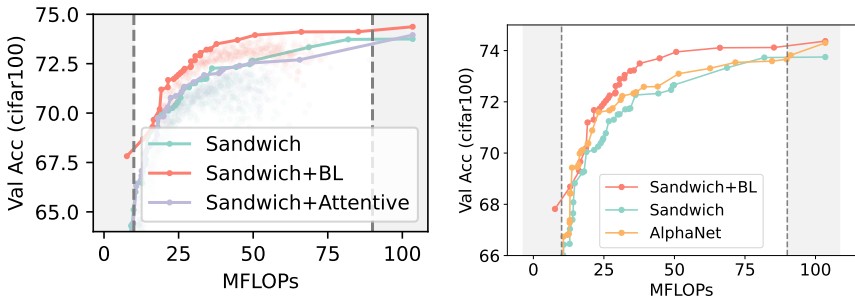

Figure 13: Comparison of our proposed method against AttentiveNAS (**left**) and AlphaNet (**right**) in CIFAR-100 in the MobileNetv2 search space.

Fig. 12: the improvement of boundary learning is evident right from the beginning of the search, thereby yielding an *anytime* performance benefit during the second stage of the NAS pipeline.

**Comparison against additional methods.** In this section, we include some preliminary experiments against additional methods, including AttentiveNAS Wang et al. (2021b) and AlphaNet Wang et al. (2021a). We show the results in Fig. 13: in the dataset and search space combination we considered, we found AttentiveNAS to only lead to very marginal improvement over the baseline sandwich sampling strategy. We hypothesize that a possible reason for the marginal gain of AttentiveNAS (Fig. 13(a)), in this case, is due to the fact that while AttentiveNAS aims to adapt the search space by focusing on the promising sub-networks seen so far, for a large and complex enough search space, even achieving that is difficult given the extremely large of possible candidate sub-networks. On the other hand, we find AlphaNet (Fig. 13(b)) to improve the baseline strategy over the entire Pareto front, but BL nevertheless led to a greater extent of improvement. In both cases, we find boundary learning to lead to a larger gain compared to these baselines.

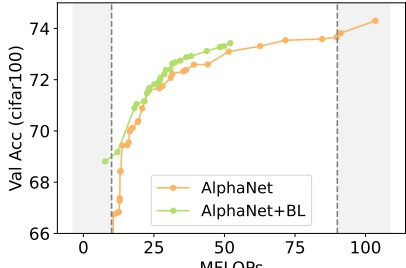

Figure 14: Comparison of AlphaNet vs AlphaNet + BL of CIFAR-100 on MobileNetv2 search space. It is obvious that BL further improves upon AlphaNet.

Additionally, as we discussed in §5 since BL uniquely targets a different aspect of the supernet training pipeline (the min and the max architectures) than the existing techniques discussed, we argue BL may be used *in combination* with them. As a proof of concept, we also show a preliminary result when BL is orthogonally applied on top of AlphaNet in Fig. 14, a method that we observed to lead to consistent improvement over baseline sandwich sampling – we defer a thorough investigation to future works.

## C    Ablation Studies

**Learning the min only vs. learning both max and min.** Our method is compatible with the setting where only the min or max network is learned, and the other extremum is fixed. This could be useful when partial prior knowledge about the search space exists. The simplification to accommodate the case where only a single extremum is learned is straightforward: instead of learning two policies, we only learn one, and we also adjust the sampling strategies for the regular random architectures accordingly. Given that the results in the main text §4.3 suggests that the min network is more important for the overall training, we conduct comparisons if only min is learned and max is fixed at the ground-truth largest network, and we show the results in Fig. 15. We find that in both cases, learning both min and max still outperforms min-only, even though in the MobileNetv2-like search space, the policy eventually converges to the largest

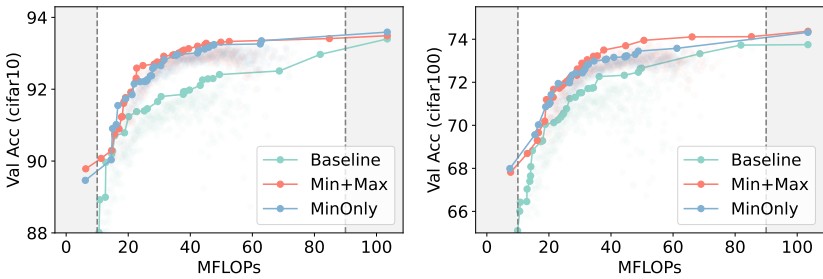

Figure 15: Comparison of the Pareto fronts of top-1 accuracy vs MFLOPs of architectures in the MobileNetv2-like search spaces in CIFAR-10 and CIFAR-100. `Min+Max` denotes boundary learning with both min and max networks learned (identical to the results presented in the main text). `MinOnly` denotes the case where only min network is learned.

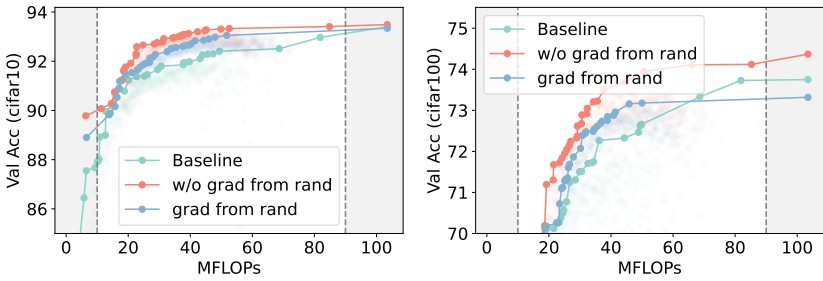

Figure 16: Comparison of the Pareto fronts of top-1 accuracy vs. MFLOPs of architectures in the MobileNetv2-like search spaces in CIFAR-10 and CIFAR-100. `w/o grad from rand` denotes the boundary learning with gradients to the policy updated from min and max networks only (identical to the results presented in the main text). `grad from rand` denotes the case where random architecture gradients also update the policy.

network in the search space (Fig. 6 to 8 in the main text). Finally, we find that learning the min only already leads to a large improvement over the vanilla sandwich sampling baseline without boundary learning.

**Effect of back-propagating gradients from random architectures.** We also investigate a variant of the boundary learning algorithm presented in Algorithm 1 in the main text, but we also update the gradients of the policy using random architectures. Recall that in §3, we retain the uniform sampling strategy for the regular, random architectures but condition on the max and min architectures learned, and we use a Heaviside step function to determine the candidates to be included on each search dimension for the sampling. By using a straight-through estimator over the hard Heaviside step (i.e., we retain the hard Heaviside step for the forward backpropagation but use a hard sigmoid for back-propagation), we may retain the differentiability and allow gradients to be passed from the random architectures as well – the advantage of this approach is that it allows the policy to

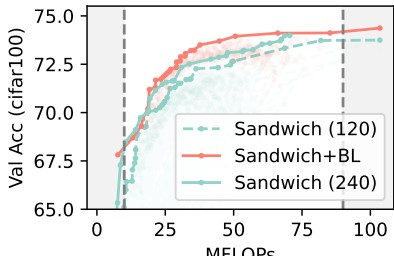

Figure 17: Comparison against baseline supernet training with 240 epochs in MobileNetv2-like space (§4.2) on CIFAR-100

be updated more frequently, and in cases such as large-batch training where the number of gradient updates per epoch is small, the policies may converge faster than otherwise. We present the results in Fig. 16. We found that although using the additional gradient information increases the convergence speed of the policy, we obtain better Pareto fronts by using the gradients from min and max networks only, likely because the random network gradients are noisy and may conflict with the gradients from max and/or min. Nonetheless,

techniques to reduce gradient variance or to remedy gradient conflicts may be used; we defer a thorough investigation to future work.

**Comparing against longer training.** To further demonstrate the effectiveness of boundary learning, we also compare against baseline supernet training with a doubled number of training epochs (240), and we show the results in Fig. 17: while longer training indeed improves the supernet performance, it is still outperformed by boundary learning, which only trains for 120 epochs and is thus approximately half as expensive.

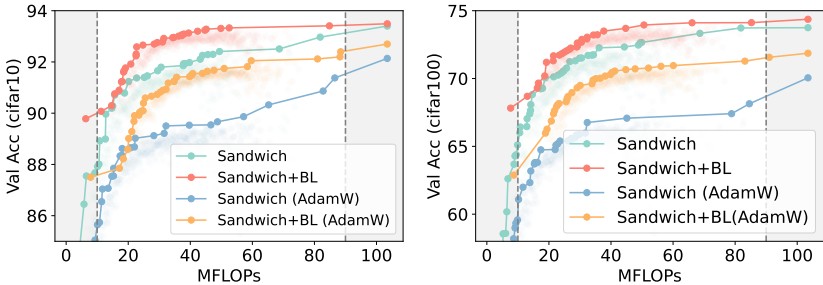

Figure 18: Comparison between baseline and BL in CIFAR-10 (**left**) and CIFAR-100 (**right**) of the MobileNetv2 search space when AdamW is used instead. It is evident that using AdamW leads to worse results in general, but BL still improves significantly over the baseline.

**Comparing against alternative optimizer choices.** We have largely followed the supernet training protocol from previous works and adopted the standard SGD with momentum optimizer. In this paragraph, we investigate the robustness of the empirical gain of BL by using an alternative optimizer. As we show in Fig. 18, we run both the baseline sandwich sampling and BL with an AdamW optimizer (Loshchilov & Hutter, 2018), which is essentially Adam (Kingma & Ba, 2015) with a decoupled weight decay – we modified the max learning rate to $10^{-3}$ and the weight decay to 0.1 in AdamW, and left all other hyperparameters, including the learning rate schedule, unchanged. We find that while using AdamW leads to some performance deterioration compared to the main results with SGD, the extent of deterioration is roughly the same for baseline and BL, meaning that the extent of gain of BL over the baseline in a relative scale is largely unchanged– the experiment demonstrates the importance of hyperparameter selection in NAS, but at the same time also confirms the robustness of empirical gain delivered by BL even when a sub-optimal hyperparameter setting is applied.

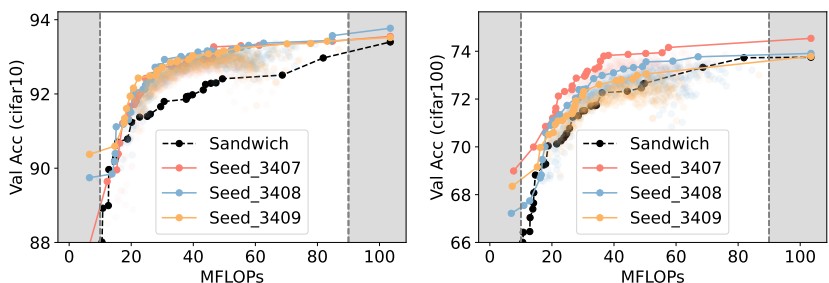

Figure 19: Effect of random seeds on the performance of BL in CIFAR-10 (**left**) and CIFAR-100 (**right**) of the MobileNetv2 search space. `Sandwich` denotes the baseline sandwich sampling training.

| Dataset | Sandwich | BL | | | Average±std |
|---------|----------|------|------|------|-------------|
| Seed | | *3407* | *3408* | *3409* | |
| CIFAR-10 | 1423.18 | 1473.50 | 1490.86 | 1487.48 | 1483.95±7.51 |
| CIFAR-100 | 2366.57 | 2464.03 | 2412.13 | 2369.74 | 2415.30±38.6 |

Table 4: Hypervolumes of Pareto fronts of BL under different seeds vs. baseline sandwich training. The reference point for CIFAR-10 is set at [79.0, -113.7] (Accuracy, MFLOPs), and is set at [50.1, 113.7] for CIFAR-100.

**Effect of random seeds.** In this section, we run the proposed algorithm with multiple random seeds to validate its robustness, and we demonstrate the results in Fig. 19, where we compare the Pareto front obtained in different seeds and their comparison against the baseline sandwich sampling, and in Table 4, where we quantitatively present the hypervolume in each case. We find that our proposed BL is largely robust to the randomness induced by different seeds, and the performance is stable, consistently outperforming the baselines.

