# OpenReview forum: "Learning Search Space Boundaries Improves Supernet Training"
_TMLR — Rejected by TMLR_

### Review · Reviewer_m5AY · 2023-08-29

**Summary Of Contributions:**

This paper proposes a simple yet effective modification to the supernet training methods, where a policy that defines the search space is learned jointly with supernet weights in an end-to-end, differentiable manner during supernet training. Starting with completely uninformative priors about the search space at initialization, the policies jointly learn a subspace within the original broad search space. The proposed approach incurs negligible additional computational overhead, requires no further retraining or fine-tuning.

The experiments show that the approach consistently discovers reasonable search space boundaries in huge, realistic search spaces where even the current state-of-the-art methods fail and yields high-performing supernets.

**Audience:**

Yes

**Claims And Evidence:**

No

**Requested Changes:**

See the weakness.

The claim about addressing the search space issue may not be accurate. It is better to provide more discussions or experiments.

The current writing mainly focus on the detailed method and its performance, without insights on why it can work better or the mechanisms of the method. It is better to discuss the reason why the proposed method can perform better.

The experiments can be enhanced. The current results demonstrate that the improvements over sandwich sampling perform better than the original sandwich sampling. But without detailed comparison with other kinds of supernet training methods or baselines, we are still not sure whether the proposed method can perform better. It is better to compare with more baselines.

**Strengths And Weaknesses:**

\+ It performs a detailed analysis about the learned space and shows that the min network has a larger influence on the overall supernet training. It further performs ablation study on sampling min network.

\- It discusses the limitations of the current NAS works on the search space. Many works rely on carefully engineered search spaces with expert domain knowledge. Then it claims to address the search space issue. The claim may not be very accurate. (1) The proposed method mainly improves the supernet training, without detailed design on the search dimension. (2) The experiments mentions the search space design, such as including the search for the activation function ({ReLU, swish}) and whether to use squeeze-and-excite (SE) module for MobileNetv3-like search space. The design seems to be heuristic and still heavily rely on human experts. It does not seem to address the search space issue. (3) It claims that 'our search space is far more generalizable than previous bespoke search spaces defined using prior knowledge'. But since it compares with the baseline on the same search space in experiments, we are not sure whether the search space in the paper performs better, as we do not have baselines on search space. All comparisons are conducted on the same search space, we are not sure whether the search space is better. (4) Some search space design mainly follows previous works, such as Chin et al. (2021) in Width-only Search Space, Wang et al., 2021b;a in MobileNetv2-like search space. It is hard to understand what are the differences between the search space in the paper and previous works. It is better to make it more clear and specific on the differences compared with specific works. The claim seems not to be well supported and I am confused with the search space issue.

\- Starting with completely uninformative priors about the search space at initialization, the method learns a subspace within the original broad search space and performs better, following the sandwich sampling method. It is not clear why searching in a larger space does not achieve better than searching within a smaller subspace. Since the larger space covers the smaller subspace, it usually has a solution not worse than that in the subspace. The current experiments only show results in figures, without detailed discussion or analysis about this point. It is better to discuss the reason why the proposed method can perform better. The current writing mainly focus on the detailed method and its performance, without insights on why it can work better or the mechanisms of the method.

\- The experiments can be enhanced. It mainly compares with the original sandwich sampling method (Yu et al., 2019) as it follows the sandwich sampling framework and propose improvements on that. It currently only has one baseline which is released 2019. It is better to compare with more NAS baselines to conduct a comprehensive comparison. For example, (Wang et al., 2021b) explores NAS with Attentive Sampling. (Wang et al., 2021a) improves supernet training with Alpha-Divergence. They may have some search space engineered by human experts. But these methods are mainly related to supernet training techniques rather than search space design, and it is easy to conduct the comparison in the same search space. The current results demonstrate that the improvements over sandwich sampling perform better than the original sandwich sampling. But without detailed comparison with other kinds of supernet training methods or baselines, we are still not sure whether the proposed method can perform better. It is better to compare with more baselines.

---

> ### Author Response · Authors · 2023-10-05
> **Response (Part 1)**
>
> We thank the reviewer for their detailed and constructive feedback. We'd like to refer the reviewer to both our paper and our response below.
>
> > The experiments mentions the search space design, such as including the search for the activation function and whether to use squeeze-and-excite module for MobileNetv3-like search space. The design seems to be heuristic and still heavily rely on human experts. It does not seem to address the search space issue.
>
> While we agree the decisions on whether to search for squeeze-and-excite and activation functions require human input, we argue that human-supplied building blocks or search dimensions are at the foundation of *any* NAS method. Including these as search dimensions creates a more flexible and less hand-crafted search space. In contrast,  previous papers using the MobileNetv3-like search spaces (e.g., see Table 7, Appendix B of [Wang et al. 2021a]) have chosen to hard-code activations and SE module choices at specific layers (whether to enable or disable them) in a manual and opaque way, without explanation. By incorporating them as search dimensions rather than manually fixed designs that are searched via a NAS algorithm in an end-to-end manner, we have already significantly increased the extent of automation.
>
> We have added explanations to make the above point clearer in Sec 4.2.
>
> > It claims that 'our search space is far more generalizable than previous bespoke search spaces defined using prior knowledge'. But since it compares with the baseline on the same search space in experiments, we are not sure whether the search space in the paper performs better, as we do not have baselines on search space. All comparisons are conducted on the same search space, we are not sure whether the search space is better.
>
> We performed all experiments in the same search space, as the critical difference w.r.t the baseline is that our method adapts the search space with all other training recipes unchanged, and thus experiments in the same space allow us to attribute any performance difference to boundary learning, without other confounding variables. While it is possible to include baselines using alternative search spaces, such as the highly engineered search spaces in previous works [Wang et al. 2021a and 2021b], we opt not to do so for the following reasons:
>
> Firstly, these heavily human-engineered search spaces often feature hard-coded yet performance-sensitive* (see *Note* below) design options (e.g., the depth and width of certain layers are fixed) that are likely the result of laborious iterations on the search spaces. On the other hand, our method requires no iteration at all, with search space adaptation taking place in an end-to-end, one-shot manner.
>
> Secondly, experiments on these alternative search spaces would mean that supernet training would *start* on a search space with human-engineered boundaries, making search space adaptation less critical. Our method starts with a generic search space and faces a more challenging task of optimizing supernets *while* figuring out the search space boundaries. It is worth noting that, as we argued right from the introduction (e.g., page 2), our method aims to address the insufficiency of existing supernet-based NAS methods in a practical setting where human knowledge of the search space is unavailable (e.g., for a disparate, novel task). If, however, extensive human knowledge is already available from expensive iterations or otherwise (as represented by the baseline in engineered search spaces), one can be indeed better off using a search space configuration inspired by these knowledge rather than using on-the-fly boundary learning (which, as we mentioned, entails a more difficult task) – this is, however, a setup already extensively studied and not one that we aim to address.
>
> *Note*: We show this with the poor performance of the baseline supernet training in a generic search space (defined in Table 2) generated from the original MobileNet configuration with a simple formula without fixing or otherwise customizing some layers.

---

> > ### Author Response · Authors · 2023-10-05
> > **Response (Part 2)**
> >
> > > Some search space design mainly follows previous works,... It is hard to understand what are the differences between the search space in the paper and previous works. It is better to make it more clear and specific on the differences compared with specific works.
> >
> > While we developed the search space from previous works, we built generic search spaces straightforwardly expanded from the standard MobileNet architectures, *without informed priors*. This differs from previous works which designed layer-specific boundaries as seen in (cf, Table 7 in App. B of [Wang et al. 2021a], where, the depth search dimension differs from layer to layer—with deeper layers given higher minimum and maximum depths to ensure shallow networks are not included. Furthermore, SE modules and activation per layer are hand-picked and fixed.
> >
> > The difference may seem subtle, but it is critical: we show that tailored search space design—which requires a massive amount of human iteration and is likely task-specific—is critical to the performance of the supernet, as the baseline supernet training performance otherwise degrades significantly in generic search spaces (shown in Fig 2). This motivates our method.
> >
> > > It is not clear why searching in a larger space does not achieve better than searching within a smaller subspace. Since the larger space covers the smaller subspace, it usually has a solution not worse than that in the subspace.
> >
> > While the reviewer is right in saying that a larger search space contains a smaller subspace, in supernet-based NAS methods, larger search spaces do not necessarily outperform in practice: supernets aim to train *all* networks in the search space simultaneously, and as the search space becomes larger, the networks contained in it become more different, and it becomes increasingly difficult to improve all network weights simultaneously. Both of these phenomena, as well as the benefit of using smaller subspaces, are well-documented. For example, [Zhao et al. 2021] attempt to divide the search space into multiple subspaces and train a supernet in each to alleviate supernet training difficulties in large search spaces. These points have been highlighted in Sec 3, “Limitations of current methods.”
> >
> > > The current experiments only show results in figures, without detailed discussion or analysis about this point. It is better to discuss the reason why the proposed method can perform better. The current writing mainly focus on the detailed method and its performance, without insights on why it can work better or the mechanisms of the method.
> >
> > We agree with the reviewer that more high-level intuition of the working mechanism would improve the presentation, and we’ve expanded our discussion at the beginning of Sec 3, and we outline our response to this point below:
> >
> > Related to the response above, the key mechanism is that our method performs on-the-fly search space adaptation. Starting from a large, generic search space without informed priors, our method gradually adjusts the boundaries by excluding harmful operators from the search space (e.g., networks with very small depths). This reduces the search space complexity (hence reducing the conflicts amongst the sub-networks) and allows more performant supernet training.
> >
> > It is worth noting that given the weak performance of generic search space (as shown in Fig 2), prior works likely arrived at the final search space design with per-layer customized boundaries by adapting boundaries in a trial-and-error manner. However, crucially, our method requires no such expensive manual iterations and achieves adaptation in a single supernet training.
> >
> >
> > > The current results demonstrate that the improvements over sandwich sampling perform better than the original sandwich sampling. But without detailed comparison with other kinds of supernet training methods or baselines, we are still not sure whether the proposed method can perform better. It is better to compare with more baselines.
> >
> > We added some comparisons with [Wang et al. 2021a and 2021b] in the App. B. While these methods led to some improvements over the baseline, they are still outperformed by boundary learning. This is consistent with various previous findings that highlighted the importance of search space design in NAS.
> >
> > Furthermore, it is worth noting that our method targets a fundamentally different aspect compared to these methods. For example, as we discussed in Sec 5, [Wang et al. 2021a] target the sampling strategy of the random architectures (using the terminology in Eq. (3)), whereas [Wang et al. 2021b] target the loss function design. As such, the methods are fully orthogonal to each other, and we indeed expect some combinable benefits. We added a preliminary experiment in the App. B showing this and defer a thorough investigation to future work (we added an explanation in the Conclusion).

---

### Review · Reviewer_SbUg · 2023-09-09

**Summary Of Contributions:**

This paper proposes to learn a policy that starts with a search space and refines it. The policy is learned along with the search space. The policy adapts the maximum and minimum as the supernet is being trained. The proposed approach has been evaluated on multiple datasets, and in all cases, the proposed solution improves the trade-off front over a baseline that does not adapt the search space.

**Audience:**

Yes

**Broader Impact Concerns:**

There are no ethical implications for this work, so there is no need for a Broader Impact Statement.

**Claims And Evidence:**

No

**Requested Changes:**

- Other existing methods adapt the search space while training the supernet, like what is being done in this paper. Examples include [1-2]. Please discuss (qualitatively and quantitatively) the similarities and differences of the approach in the current paper with these types of approaches.
- The paper's tone in the introduction suggests a general search space design, including expanding and shrinking the spaces. However, ultimately, the proposed method can only shrink a pre-existing search space. So, the claims are more modest than suggested by the title and introduction. Please rewrite both to more accurately reflect the capabilities of the approach in the paper.
- The search spaces considered here are discrete, with only a few choices for each. Could you comment on how well the policy learning would work for, say, a continuous space or a discrete space with a very large cardinality?
- It would be good to see how efficient the search is in terms of the number of architectures evaluated between the baseline and the proposed method to reach a given accuracy value.
- I believe the training setup in terms of choice of optimizer and other training hyper-parameters has been fixed. Would the observations being made in terms of the benefits afforded by adaptable search translate to the same extent if one were to use other optimizers (say Adam instead of SGD for supernet weights) and other choices for training, such as using better data augmentation strategies, longer epochs (this is already done), etc.?

**Questions:**

- The motivation stems from the observations in Figure 2, where 30 sub-networks were sampled from each of the search spaces considered. This experiment is biased towards small search spaces, independent of the supernet training. Optimal architectures are probably concentrated in a few high-density regions of the search space. The coverage of such modes is exponentially less likely (assuming the sampling is random) as you increase the dimensionality of the search space. So, a fixed set of 30 random samples is unlikely to hit good sub-networks for large spaces. In light of this, it is unclear how reliable the observations in Fig. 2 are. Can you comment on this?
- The accuracies on some datasets, like CIFAR-10 and CIFAR-100, are lower than what can be achieved currently with 32x32 images. I suspect it is due to the lack of data augmentation or not using better optimizers. Could you comment on this?

**Other Comments:**

- On page 8, in the settings paragraph, possible typo. "After the supernet training, we closely follow Wang et al. (2021b) and Wang et al. (2021b) to run an adapted version of genetic algorithm...". It should probably be Wang et al. (2021a) and Wang et al. (2021b).
-

[1] NSGANetV2:Evolutionary Multi-Objective Surrogate-Assisted Neural Architecture Search, ECCV 2020

[2] Neural Architecture Transfer, IEEE Transactions on Pattern Analysis and Machine Intelligence 2021

**Strengths And Weaknesses:**

- Strengths
  - Designing better search spaces is a long-standing problem in neural architecture search, with relatively not much attention from the research community. So, it is refreshing to see a paper try to address this problem.
  - The results show that adapting the search spaces yields appreciable improvements in the trade-off fronts. This is not new, though. The same observation was made by existing work that also adapts the search space during the search process.
- Weaknesses
  - While the method clearly shows the benefit of adapting the search spaces, comparisons (qualitative and quantitative) to other search space adaptation methods are lacking. This would help put the current paper in the context of other attempts at search space design.
  - The paper's tone in the introduction suggests a general search space design, including expanding and shrinking the spaces. However, ultimately, the proposed method can only shrink a pre-existing search space. So, the claims are more modest than suggested by the title and introduction.

---

> ### Author Response · Authors · 2023-10-05
> **Response (Part 1)**
>
> We thank the reviewer for their detailed and insightful comments. Please see below for our response.
>
> > While the method clearly shows the benefit of adapting the search spaces, comparisons (qualitative and quantitative) to other search space adaptation methods are lacking. This would help put the current paper in the context of other attempts at search space design.
>
> > Other existing methods adapt the search space while training the supernet, like what is being done in this paper. Examples include [1-2]. Please discuss (qualitatively and quantitatively) the similarities and differences of the approach in the current paper with these types of approaches.
>
> We thank the reviewer for suggesting these related works. We include the requested discussion below, which has also been added to Sec 5 in the paper.
>
> Both NAT [1] and NSGANetV2 [2]  focus on the fast adaptation of neural architectures for diverse tasks and/or many conflicting objectives. The key part where both NAT and NSGANetV2 bear a resemblance to our method is Sec 3.6, Supernet Adaptation for NAT, and Sec 3.3, Speeding Up Upper-Level Optimization in NSGANetV2, where the authors proposed to focus on the promising subnetworks recommended by the evolutionary search algorithm so far instead of uniformly sampling from all possible subnetworks. The key similarity here is that both our method and these works propose to alter the distribution during supernet training, but the similarity ends here: in our view, the adaptation strategy in NAT/NSGANetV2 is more similar in both motivation and execution to AttentiveNAS and Joslim (which we discussed in the paper): these papers propose to sample near the Pareto front (i.e., the more promising architectures so far) for the random architectures (using our notation in Eq. (3) in our paper) using accuracy predictor or an alternative mechanism.
>
> These methods, however, do not necessarily address the search space issue: if sandwich sampling is used, while the random architectures are now sampled according to an estimated distribution of optimal subnets (e.g., Fig 19 of NAT), the max and min networks would still be fixed at the largest and smallest networks, respectively. As seen in our response to Reviewer m5AY, we compare BL to AttentiveNAS (which is conceptually similar to the methods mentioned by the reviewer) in new experiments added to the App. B, we found that adapting max and min networks (as done by BL) has a much larger performance impact. However, as we’ve also pointed out in our response to Reviewer m5AY, we argue that these methods targeting the random architectures are fully orthogonal to our method targeting the min and max, and we view a comprehensive investigation of this as an important future work.
>
> > The paper's tone in the introduction suggests a general search space design, including expanding and shrinking the spaces. However, ultimately, the proposed method can only shrink a pre-existing search space. So, the claims are more modest than suggested by the title and introduction.
>
> We’ve mainly considered search space shrinking as existing supernet methods tend to struggle with an overly large search space rather than an overly small one (we mentioned this is our main objective in the Introduction, such as “the policies jointly learn a subspace within the original broad search spaces,” “discovers reasonable search space boundaries in huge, realistic search spaces).
>
> It is, however, worth noting that our method is perfectly compatible with search space expansion if the prior on the max network is not initialized in the way we defined in Eq. (5), where we place the most probability mass of the max policy on the largest network and the min policy on the smallest network. If one does not do so, it is indeed possible to achieve search expansion as the training proceeds. Again, we did not perform such an experiment because, for a completely uninformative search space, we believe it is the most reasonable and convenient to place the prior of max and min policy on the largest and the smallest networks, respectively, which imitates the behavior of the baseline sandwich strategy.
>
> We added a discussion of the above, as requested by the reviewer, both as a footnote in the Introduction and additional discussions in Sec 3.
>
> > The search spaces considered here are discrete, with only a few choices for each. Could you comment on how well the policy learning would work for, say, a continuous space or a discrete space with a very large cardinality?
>
> We mainly considered discrete search dimensions with a few choices as this is, by far, the most prevalent type of search space NAS problems consider. If a search dimension is continuous or can be well-approximated as continuous, we can compute gradients w.r.t. the search space boundary itself instead of the parameters that reparameterize the boundary (in our case, the parameters of the Gumbel-softmax distributions), which would simplify the optimization problem.

---

> > ### Author Response · Authors · 2023-10-05
> > **Response (Part 2)**
> >
> > > It would be good to see how efficient the search is in terms of the number of architectures evaluated between the baseline and the proposed method to reach a given accuracy value.
> >
> > We are unsure whether the reviewer meant the number of architectures during training or evaluation. If they meant the former, we did not show the training curve, as BL adjusts the configuration of the max and min networks, whereas they are static in the baseline. Thus, if we plot the two training curves, they are actually showing the performance of different subnetworks sampled during training and are hence not comparable. If they meant the latter, we’ve added a figure showing the evolution of the discovered Pareto front (accuracy and MFLOPs trade-off) of the discovered networks as the evaluation proceeds in the App. B – however, it is worth noting that we used the standard evolution search without modification from [Wang et al. 2021a,b], and any efficiency gain purely stems from the better search space from BL rather than from the search algorithm.
> >
> > > I believe the training setup in terms of choice of optimizer and other training hyper-parameters has been fixed. Would the observations being made in terms of the benefits afforded by adaptable search translate to the same extent if one were to use other optimizers (say Adam instead of SGD for supernet weights) and other choices for training, such as using better data augmentation strategies, longer epochs (this is already done), etc.?
> >
> > We’ve largely inherited the hyperparameter and data augmentation settings from previous works, as we would like to certainly attribute any performance difference to the proposed strategy instead of any confounding factors. We’ve nevertheless included an additional experiment where AdamW is used instead of SGD for supernet training, as evidenced in the additional experiments in the App. C. We observe that the benefits of BL persist.
> >
> > > The motivation stems from the observations in Figure 2, where 30 sub-networks were sampled from each of the search spaces considered. This experiment is biased towards small search spaces, independent of the supernet training. Optimal architectures are probably concentrated in a few high-density regions of the search space. The coverage of such modes is exponentially less likely (assuming the sampling is random) as you increase the dimensionality of the search space. So, a fixed set of 30 random samples is unlikely to hit good sub-networks for large spaces. In light of this, it is unclear how reliable the observations in Fig. 2 are. Can you comment on this?
> >
> > We’d like to emphasize that each box in the box plots contains 30 sampled subnetworks *within that FLOPs range* – we follow the procedure in [Wang et al. 2021b] where we sample randomly, conditional on the FLOPs range instead of sampling 30 subnetworks randomly without constraints, which would have influenced the results in the way the reviewer described. We’ve made this point clearer in the caption of Fig 2.
> >
> > > The accuracies on some datasets, like CIFAR-10 and CIFAR-100, are lower than what can be achieved currently with 32x32 images. I suspect it is due to the lack of data augmentation or not using better optimizers. Could you comment on this?
> >
> > This is likely a result of the MobileNet backbone we used — in fact, with the MobileNet backbones, our method achieves a comparable performance to previously reported open-source implementations like https://github.com/weiaicunzai/pytorch-cifar100 and https://github.com/ShowLo/MobileNetV3.
> >
> > We agree with the reviewer that better empirical performance may be obtained, most notably by using stronger backbones/search spaces and improved augmentation and training recipes as the reviewer suggested. While we deem this as important future work to be done (we included this as a limitation and future work in the Conclusion), we also argue that our method is largely search space-agnostic and should work wherever sandwich sampling works (e.g., in a transformer or hybrid search space in [Gong et al. 2022]).
> >
> > > On page 8, in the settings paragraph, possible typo. "After the supernet training, we closely follow Wang et al. (2021b) and Wang et al. (2021b) to run an adapted version of genetic algorithm...". It should probably be Wang et al. (2021a) and Wang et al. (2021b).
> >
> > We thank the reviewer – the typos have been corrected.

---

### Review · Reviewer_3Fmr · 2023-09-20

**Summary Of Contributions:**

Neural Architecture Search (NAS) automates the manual design of neural network architectures by formulating it as an optimization problem. Within a predefined search space, the objective is to identify the Pareto-optimal set of neural network architectures that simultaneously enhance performance and efficiency. In its standard form, NAS is prohibitively expensive to be practical. To overcome the prohibitive cost of multiple training runs, weight-sharing-based NAS first trains a single super-network that encompasses all potential networks within the search space. In a second step, we can select the optimal network configurations with multi-objective search where we estimate the performance of neural networks based on the shared weights of this super-network.

Prior research has demonstrated that the choice of the search space plays a major role in NAS and often dictates the ultimate outcomes. This paper introduces an adaptive mechanism that dynamically updates the lower and upper bounds of the search space. This adjustment makes Weight-sharing-based NAS less sensitive to the precise definition of the search space. The proposed method appears to be straightforward to integrate with existing NAS techniques from the literature and shows performance improvements across various mobile-net search spaces used in image classification problems.

**Audience:**

Yes

**Broader Impact Concerns:**

I don't expect any ethical implication of this work

**Claims And Evidence:**

Yes

**Requested Changes:**

- Do you also use the learned bounds in the search phase, for example for the genetic algorithm?

- Report results aggregated over multiple runs and report uncertainty bars.

- Typo caption Figure 3: '... with (Sandwich) and without (Sandwich + BL) ...' should be the other way round

- Typo under equation 3: delta^{di-1} is mentioned twice

**Strengths And Weaknesses:**

## Strengths

- *Writing*: Overall I found the paper to be well written. The central arguments of the paper are easy to follow and well motivated.

- *Approach*: I agree with the authors that neural architecture search stands and falls with a good search space. It is also true that the is not often not immediate clear how to design the bound of the search. The proposed approach represents a save guard to be more adaptive and to avoid catastrophic failure.

- *Experiments*. The empirical evaluation  seem sound overall and show a strong benefit of the proposed method. However, see also points below.


## Weaknesses

My primary concern is that Figures 3, 4, and 5 appear to show, if I am not mistaken, only the Pareto front from a single run. To arrive at more statistically robust conclusions, it would be beneficial if the same experiment could be replicated with different random seeds. Ideally, the quality of the Pareto front could be assessed, for instance, in terms of hypervolume improvement, thus enabling the inclusion of uncertainty bars.

---

> ### Author Response · Authors · 2023-10-05
>
> We thank the reviewer for their detailed feedback. Please see below for your response to the concerns raised.
>
> > My primary concern is that Figures 3, 4, and 5 appear to show, if I am not mistaken, only the Pareto front from a single run. To arrive at more statistically robust conclusions, it would be beneficial if the same experiment could be replicated with different random seeds. Ideally, the quality of the Pareto front could be assessed, for instance, in terms of hypervolume improvement, thus enabling the inclusion of uncertainty bars.
>
> > Report results aggregated over multiple runs and report uncertainty bars.
>
> We thank the reviewer for their suggestion, and we have incorporated a figure in the App. C to show multiple seeds and standard deviation measured with hypervolume. We are currently showing CIFAR-10 and CIFAR-100 on MobileNetv2 search space due to time and resource constraints, but we will endeavor to include multiple seeds for other experiments. It is worth noting, however, that the improvements both visually and in terms of hypervolume over the baseline are quite significant over different model and dataset combinations. This, in our opinion, is likely to suggest any improvement difference comes from our method instead of stochasticity.
>
> > Do you also use the learned bounds in the search phase, for example for the genetic algorithm?
>
> Yes. We have added a clarification on Page 9, Sec 4.1.
>
> > Typo caption Figure 3: '... with (Sandwich) and without (Sandwich + BL) ...' should be the other way round
>
> > Typo under equation 3: delta^{di-1} is mentioned twice
>
> We thank the reviewer, and we’ve corrected these typos.

---

### Decision · Action_Editor_HUtu · 2023-10-27

**Recommendation:** Reject

**Comment:**

All the reviewers agreed that the methodology and experiments in the paper are valuable and of interest to the TMLR community. It is clear that the method is effective in applying Sandwich Sampling to somewhat larger search spaces. I was particularly happy to see the authors follow one reviewer suggestion to perform multiple runs of some experiments; in future versions I suggest continuing with multiple runs of the baselines as well.

However, two of the three reviewers believed the paper did not meet the Claims and Evidence criteria of TMLR. To summarize these concerns: (1) The paper claims the method allows NAS to be applied to larger and more complex search spaces that lack human prior knowledge, but the search spaces in experiments still rely on extensive human knowledge that seems to far outweigh the reduction of prior knowledge in expanded boundaries (especially compared to early NAS work that did not rely on a predefined backbone or supernet); (2) The paper makes claims about the implications of the method for NAS in general, but only compares to Sandwich Sampling; (3) Missing evidence of cases when one of the larger search spaces leads to better generalization than smaller search spaces, which is a key motivation for this kind of work. The author responses mainly defended their initial claims, and although they made some helpful and intriguing points, they did not sway the reviewers on their main concerns. For example, several of the places where text was added in the paper of how the method could apply in other cases further strengthened the reviewers' view that the claimed implications are beyond what the paper shows methodologically and empirically.

Some examples from the paper and author responses that raised concerns:
- “in a completely uninformative search space” The search spaces in the paper are highly-informed.
- "our search space does not rely on informed priors from human trial-and-error which, as some previous works have shown (Tu et al., 2022), is likely task-specific and might not necessarily transfer into novel tasks." The paper does not demonstrate transfer to novel tasks.
- "on tasks without strong priors, our solution consistently discovers performant subspaces within an initially large, complex search space (where even the state-of-the-art methods underperform)" There are still strong priors, and state-of-the-art methods are not tested in these newly defined spaces.

The issues with the claims are woven into the story throughout the paper, so I don't think it would be sufficient for the authors to independently address each place of concern; rather I agree with Reviewer SbUg that the high-level presentation should be adjusted, and a major revision is needed that will take some thought and time. With such a revision of the story and claims, this paper could serve as solid work for others to build upon.

Overall, the reviewers agreed that the search space design problem for NAS is important, and the method in the paper takes a step to address an aspect of the problem, but the claims exceed the method and experiments presented.

**Audience:**

Yes.

**Claims And Evidence:**

No. The claims of the generality and significance of the problem being solved are too strong, given that the method is developed from a specific flavor of NAS (sandwich sampling), quantitative comparisons are only to this method, and the search space expansions are modest compared to the prior knowledge in the design of the supernet.

**Resubmission Of Major Revision:**

The authors may consider submitting a major revision at a later time.